# Rare genetic variants affecting urine metabolite levels link population variation to inborn errors of metabolism

Yurong Cheng[1,2], Pascal Schlosser [1], Johannes Hertel[3,4], Peggy Sekula [1], Peter J. Oefner[5], Ute Spiekerkoetter[6], Johanna Mielke[7], Daniel F. Freitag [7], Miriam Schmidts [6], GCKD Investigators*, Florian Kronenberg [8], Kai-Uwe Eckardt [9,10], Ines Thiele[3,11,12], Yong Li [1] & Anna Köttgen [1,13]✉

Metabolite levels in urine may provide insights into genetic mechanisms shaping their related pathways. We therefore investigate the cumulative contribution of rare, exonic genetic variants on urine levels of 1487 metabolites and 53,714 metabolite ratios among 4864 GCKD study participants. Here we report the detection of 128 significant associations involving 30 unique genes, 16 of which are known to underlie inborn errors of metabolism. The 30 genes are strongly enriched for shared expression in liver and kidney (odds ratio = 65, p-FDR = 3e−7), with hepatocytes and proximal tubule cells as driving cell types. Use of UK Biobank whole-exome sequencing data links genes to diseases connected to the identified metabolites. In silico constraint-based modeling of gene knockouts in a virtual whole-body, organ-resolved metabolic human correctly predicts the observed direction of metabolite changes, highlighting the potential of linking population genetics to modeling. Our study implicates candidate variants and genes for inborn errors of metabolism.

[1] Institute of Genetic Epidemiology, Faculty of Medicine and Medical Center - University of Freiburg, Freiburg, Germany. [2] Faculty of Biology, University of Freiburg, Freiburg, Germany. [3] School of Medicine, National University of Ireland, Galway, University Road, Galway, Ireland. [4] University of Greifswald, University Medicine Greifswald, Department of Psychiatry and Psychotherapy, Greifswald, Germany. [5] Institute of Functional Genomics, University of Regensburg, Regensburg, Germany. [6] Department of General Pediatrics and Adolescent Medicine, Medical Center and Faculty of Medicine - University of Freiburg, Freiburg, Germany. [7] Bayer AG, Division Pharmaceuticals, Open Innovation & Digital Technologies, Wuppertal, Germany. [8] Institute of Genetic Epidemiology, Department of Genetics and Pharmacology, Medical University of Innsbruck, Innsbruck, Austria. [9] Department of Nephrology and Hypertension, University of Erlangen-Nürnberg, Erlangen, Germany. [10] Department of Nephrology and Medical Intensive Care, Charité - Universitätsmedizin Berlin, Berlin, Germany. [11] Division of Microbiology, National University of Ireland, Galway, University Road, Galway, Ireland. [12] APC Microbiome Ireland, Galway, Ireland. [13] CIBSS – Centre for Integrative Biological Signalling Studies, Albert-Ludwigs-Universität Freiburg, Freiburg, Germany. *A list of authors and their affiliations appears at the end of the paper. ✉email: anna.koettgen@uniklinik-freiburg.de

Metabolites are small molecules that represent precursors, intermediates, and end-products of metabolic reactions. Their concentrations in biospecimens, such as blood or urine, are determined by their uptake, generation, breakdown, and excretion influenced by the joint actions of specific enzymes and transporters distributed across tissues and organs. Comprehensive databases of metabolic reactions in humans have been established over the past decades[1–4]. Still, our knowledge of ongoing metabolic reactions in humans is far from complete, partly owing to the limited resolution of current metabolite profiling technologies and to the limited translation of findings from cell culture and model organisms as readouts of in vivo processes in humans.

Generation of a comprehensive understanding of metabolic reactions in humans is important for several reasons: first, altered metabolite levels can cause clinical symptoms and diseases, as exemplified by elevated serum urate levels and gout and by many inborn errors of metabolism (IEMs)[5,6]. Such metabolites represent attractive therapeutic targets to prevent and treat these diseases. Second, emerging evidence implicates metabolites as important second messengers in inter-organ communication and in signaling processes[7,8]. Insights into metabolite handling may therefore advance our understanding of human metabolism on a systemic level, beyond the tissues directly involved in their absorption, generation, metabolism, and excretion. Third, metabolites are commonly used to diagnose, stage, and monitor disease, such as creatinine levels for chronic kidney disease (CKD)[9,10], or various amino acids and acylcarnitines in newborn screening programs worldwide. Such biomarkers may indicate pathways and/or cell types that are altered in the disease state, even if the metabolites are not causal for the disease itself[11,12].

Genetic studies of metabolites are a useful tool to expand our understanding of human metabolism: metabolite concentrations and profiles have a strong genetic predisposition[13–17]. The ability to carry out genome-wide screens allows us to identify, for each metabolite, genetic determinants of its concentration in an unbiased manner. Genetic studies can reveal both common genetic variants that influence metabolite concentrations within the physiological range, as well as rare deleterious variants that have large effects on metabolite levels and may cause recessively inherited IEMs when present in the homozygous state[18].

Most previous studies have focused on the study of common genetic variants and blood metabolite levels in the general population[13,15,19,20]. Fewer studies have investigated common genetic variants and urine metabolite concentrations[14,16,17,21]. Only two previous studies systematically evaluated the influence of rare genetic variants on the blood metabolome[18,19], and one previous study examined their effects on the urine metabolome in a smaller, population-based sample[22]. We and others have previously shown that some metabolic pathways are only uncovered in a specific biospecimen such as urine. In addition, urine metabolite concentrations provide an integrative readout of metabolic reactions across tissues and organs[17]. The study of CKD patients detects genetic effects that usually generalize to population-based samples, but has the added advantage that metabolites normally below the level of quantification can be studied, and that processes related to active tubular transport may be upregulated and hence easier to detect[17].

Here, we therefore perform a rare variant association study of the concentrations of 1487 metabolites and of 53,714 metabolite ratios in urine from up to 4864 CKD patients. We detect 128 significant associations involving 30 unique genes, 16 of which are known to underlie IEMs. We connect the genes to cell types in which they are highly expressed, the genetic variants to health outcomes in a large population study, and validate our findings via in silico modeling of gene knockouts in a virtual whole-body, organ-resolved metabolic human. Our study highlights the potential of linking population genetics to modeling to gain insights into genetic determinants of metabolite handling in humans.

## Results

The workflow in Fig. 1 provides an overview of the study design. Details about each metabolite, such as mass and pathway membership, as well as quality control metrics are provided in Supplementary Data 1. Main characteristics of the study population are shown in Supplementary Data 2: mean age was 60 years, 60% of participants were men, the mean estimated glomerular filtration rate (eGFR) was 49.5 ml/min/1.73 m$^2$, and the median urinary albumin-to-creatinine ratio (UACR) was 50.1 mg/g.

**Rare variant aggregation tests identify significant associations with the urine levels of metabolites and their ratios.** We carried out two types of rare variant aggregation tests for each of up to 11,587 genes per metabolite (ratio): a burden test, which is more sensitive when all qualifying genetic variants influence metabolite levels in the same direction, and a SKAT test, which is better suited to detect associations in the presence of variants with opposing directions (Methods). In the analyses of the individual metabolites, we detected 43 significant ($p < 1.46e{-}9$; $0.05/$ (11,552 genes*1487 metabolites*2 tests)) gene-metabolite associations with the burden test, and 48 with the SKAT test (Table 1). Of these, 38 significant associations between 18 unique genes and 37 unique metabolites were detected with both tests, while 5 associations were unique to the burden test and 10 to the SKAT test (in total: 53 associations, 26 unique genes, 51 unique metabolites; Table 1). The smallest $p$-value in both-tests was detected for the association of rare variants in *UPB1* with 3-ureidopropionate concentrations ($p$-burden $= 7.3e{-}39$, $p$-SKAT $= 3.1e{-}44$). The maximum number of qualifying variants was 25 for *HAL* (total minor allele count [MAC] of 188, $p$-burden $= 1.5e{-}11$ for trans-urocanate), and the highest total MAC was 320 for 12 variants in *ALDH9A1* ($p$-SKAT $= 7.5e{-}29$ for X-24807, a yet unnamed metabolite) (Supplementary Data 3). Aldehyde dehydrogenases detoxify aldehydes from different metabolic reactions to the corresponding carboxylic acids, with ALDH9A1 showing wide substrate specificity to amino-, aliphatic and aromatic aldehydes[23]. Additional information about the unknown molecules in Table 1 can be obtained from mzML files available through the EMBL-EBI database Metabolights (Data Availability). Supplementary Data 4 contains the 180 gene-metabolite association results that meet a suggestive significance level in one or both tests ($p < 2.2e{-}6$; $0.05/(11,552$ genes*2 tests)). Although not statistically significant after multiple testing correction, these results are likely to contain true positive findings based on biological plausibility and previous studies, such as the association of *ALPL* with phosphate-containing metabolites (minimum $p = 2.3e{-}9$)[24], *DPYD* and thymine ($p = 3.1e{-}8$)[25], *NAT1* and *NAT8* with N-acetylated metabolites (minimum $p = 1.6e{-}9$)[26,27], and *SLC10A2* with the bile acid conjugate glycochenodeoxycholate 3-sulfate ($p = 2.2e{-}7$)[28].

The analysis of urine metabolite ratios is interesting because ratios can represent readouts of enzymatic reactions or substrate exchange[13,20,29]. Thus, ratios were evaluated within and among the two super-pathways amino acids and fatty acids, which contain many metabolites actively handled along the nephron and commonly excreted in urine. Results were filtered by the $p$-gain statistic to highlight ratios that contribute information beyond their individual components (Methods). There were 68 significant ($p < 4.02e{-}11$; $0.05/(11,587$ genes*53,714 metabolite ratios*2 tests)) associations with the burden test and 67 with the SKAT test

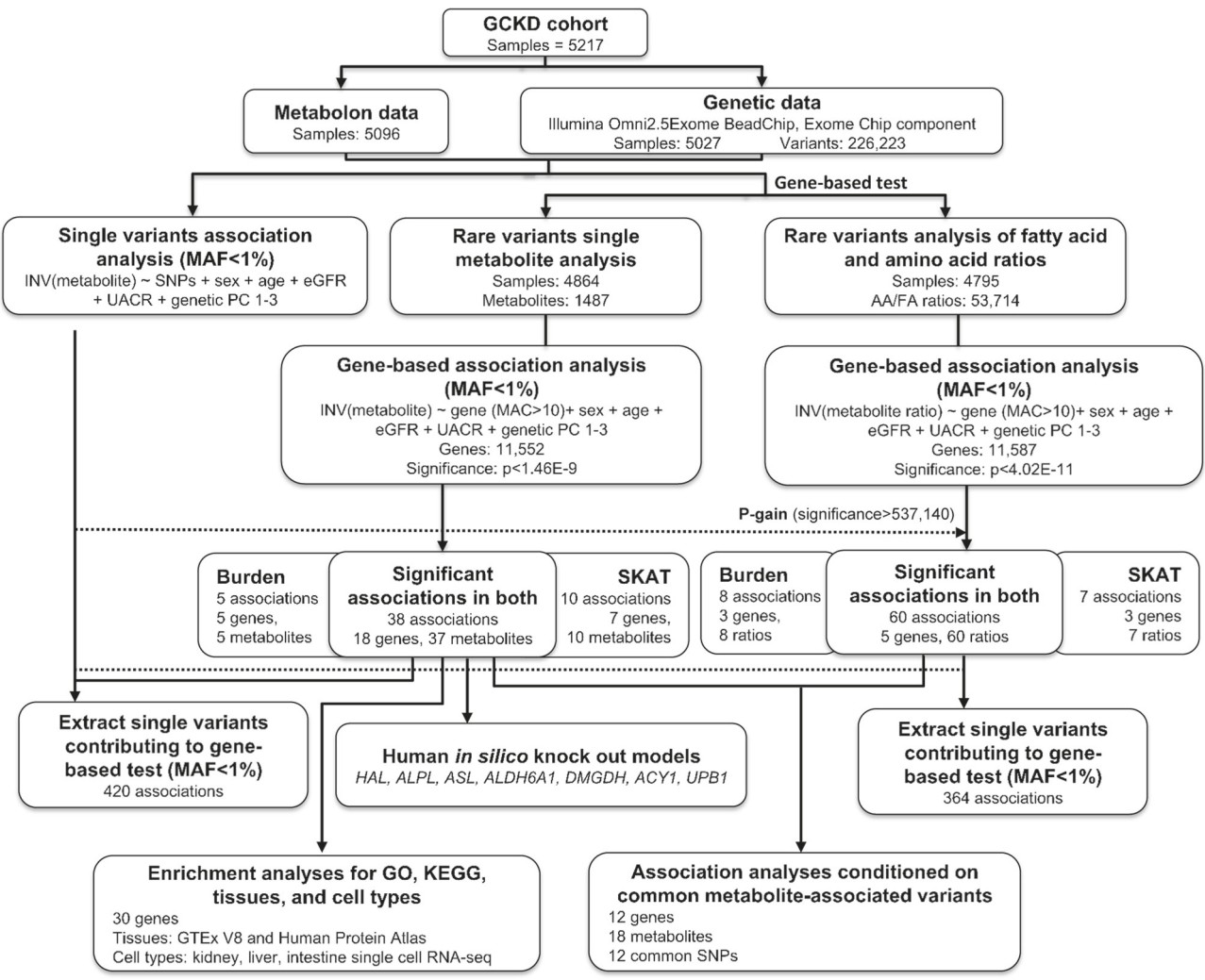

**Fig. 1 Rare variant analysis workflow.** The GCKD study enrolled 5,217 patients with moderate CKD. Non-targeted metabolite identification and quantification were conducted from urine samples using the Metabolon HD4 platform. Genotyping was performed with the Illumina Omni2.5Exome Chip. After quality control and data cleaning, genotypes of 226,233 exome chip variants and 1487 metabolites and 53,714 ratios of fatty acids and amino acids were analyzed for 4864 and 4795 patients, respectively. A burden test and the sequence kernel association test (SKAT) were carried out for each gene and each metabolite or metabolite ratio using the seqMeta R package (Methods). Carrier status of variants with minor allele frequency <1% and likely to be functional (splicing, nonsynonymous, stop gain, and stoploss) was evaluated. We used an additive genetic model and adjusted for sex, age, eGFR, UACR, and principal components. Statistical significance was defined using a Bonferroni correction and set at 1.46E−09 (single metabolites) and 4.02E-11 (ratios). Ratio results were further filtered by a p-gain of >537,140 to select ratios that carry information beyond its single metabolites (Methods). The same model was applied to obtain single variant association results for the variants included in the gene-based tests. In silico knockout models to validate findings were generated in a Virtual Metabolic Human. Enrichment analyses of significant genes were carried out using GO terms, KEGG pathways, and gene expression data from tissues and cell types. Conditional analyses were carried out to assess the effect of nearby common metabolite-associated variants on the findings from this study. MAF: minor allele frequency.

(Supplementary Data 5). Of the 7 unique genes, the associated ratios with the smallest p-values and largest p-gain in the burden or SKAT tests are displayed in Table 2. Four genes were only identified when studying metabolite ratios (*CTH*, *IVD*, *PAH*, *SLC7A9*). The largest information gain from modeling ratios over individual metabolites was observed in the SKAT test for *ACADM* and in the burden test for *CTH*. Rare variants in *CTH*, encoding cystathionine gamma-lyase, were associated with multiple cystathionine-containing ratios, reflecting its biological function. The information gain from ratios is further exemplified by the phenylalanine/tyrosine ratio, which was strongly associated with rare variants in *PAH* but its individual components were not (p-burden = 4.2e−27 for the ratio and p > e−5 for its components). *PAH* encodes phenylalanine hydroxylase, the rate-limiting step in phenylalanine

catabolism responsible for the hydroxylation of phenylalanine to tyrosine. Rare mutations in *PAH* cause the IEM phenylketonuria (MIM #261600). All 88 ratios at suggestive significance are shown in Supplementary Data 5, and again are likely to contain multiple true positive findings as evidenced by the fact that the associated metabolite ratios matched the biological functions of proteins encoded by *AASS*, *ACADSB*, and *GLB1L*.

To assess the potential contribution of nearby common genetic variants on the detected associations from the aggregate analyses of rare variants, we repeated the analyses for genes for which common genetic variants in *cis* were associated with the implicated metabolites (Methods). The effect estimates showed little effect size changes (±20%) for 10 out of 12 such genes (Table 3), suggesting that rare variant associations with

**Table 1 Genes significantly associated with urine metabolite levels ($p < 1.46E-9$).**

| Metabolite | Gene | Burden | | | SKAT | No. of SNPs used | Total MAC | Cumulative MAF of SNPs | Significant test[a] |
|---|---|---|---|---|---|---|---|---|---|
| | | P-value | Effect | SE | P-value | | | | |
| 4-Methylhexanoylglutamine | ACADM | 1.1E−09 | 0.59 | 0.10 | 9.7E−11 | 6 | 109 | 1.12% | Both |
| 4-Methylhexanoylglycine | ACADM | 2.3E−09 | 0.56 | 0.09 | 1.9E−10 | 6 | 107 | 1.15% | S |
| Heptanoylglutamine | ACADM | 1.2E−11 | 0.67 | 0.10 | 1.8E−13 | 6 | 104 | 1.24% | Both |
| Hexanoylcarnitine (C6) | ACADM | 2.0E−08 | 0.56 | 0.10 | 2.9E−10 | 6 | 98 | 1.26% | S |
| Hexanoylglutamine | ACADM | 8.5E−13 | 0.69 | 0.10 | 2.0E−14 | 6 | 109 | 1.12% | Both |
| Hexanoylglycine | ACADM | 1.0E−12 | 0.66 | 0.09 | 9.4E−15 | 6 | 109 | 1.12% | Both |
| Isocaproylglutamine | ACADM | 1.3E−14 | 0.75 | 0.10 | 6.7E−16 | 6 | 109 | 1.13% | Both |
| Isocaproylglycine | ACADM | 4.3E−12 | 0.64 | 0.09 | 7.6E−13 | 6 | 109 | 1.13% | Both |
| N-octanoylglutamine | ACADM | 1.6E−15 | 0.77 | 0.10 | 1.1E−15 | 6 | 108 | 1.16% | Both |
| Octanoylcarnitine (C8) | ACADM | 2.0E−06 | 0.46 | 0.10 | 4.5E−10 | 6 | 109 | 1.13% | S |
| X - 12398 | ACADM | 4.8E−15 | 0.75 | 0.10 | 1.9E−13 | 6 | 108 | 1.14% | Both |
| N-acetyl-aspartyl-glutamate (NAAG) | ACP2 | 2.0E−14 | −0.90 | 0.12 | 5.1E−23 | 3 | 66 | 0.68% | Both |
| Ethylmalonate | ACSF3 | 3.1E−11 | 0.53 | 0.08 | 9.4E−09 | 10 | 144 | 1.48% | B |
| Methylmalonate (MMA) | ACSF3 | 1.4E−17 | 0.69 | 0.08 | 8.4E−13 | 10 | 140 | 1.45% | Both |
| 2-Butenoylglycine | ACY1 | 5.7E−14 | 0.65 | 0.09 | 5.0E−13 | 8 | 123 | 1.32% | Both |
| N-acetyl-2-aminooctanoate[b] | ACY1 | 4.4E−10 | 0.53 | 0.09 | 4.0E−09 | 8 | 127 | 1.31% | B |
| N-acetylasparagine | ACY1 | 4.1E−11 | 0.56 | 0.08 | 1.3E−10 | 8 | 127 | 1.31% | Both |
| N-acetylglutamate | ACY1 | 1.5E−20 | 0.76 | 0.08 | 9.9E−22 | 8 | 127 | 1.31% | Both |
| N-acetylglutamine | ACY1 | 1.2E−10 | 0.54 | 0.08 | 7.9E−11 | 8 | 127 | 1.31% | Both |
| N-acetylglycine | ACY1 | 2.4E−19 | 0.80 | 0.09 | 3.1E−20 | 8 | 123 | 1.42% | Both |
| N-acetylisoleucine | ACY1 | 5.8E−17 | 0.70 | 0.08 | 8.8E−16 | 8 | 126 | 1.30% | Both |
| N-acetylleucine | ACY1 | 3.5E−10 | 0.56 | 0.09 | 1.9E−10 | 8 | 125 | 1.30% | Both |
| N-acetylserine | ACY1 | 1.4E−26 | 0.92 | 0.09 | 2.0E−27 | 8 | 127 | 1.31% | Both |
| N-acetylthreonine | ACY1 | 6.1E−16 | 0.69 | 0.09 | 2.7E−18 | 8 | 126 | 1.34% | Both |
| N-acetylvaline | ACY1 | 5.2E−20 | 0.74 | 0.08 | 3.2E−19 | 8 | 127 | 1.31% | Both |
| N-formylanthranilic acid | AFMID | 1.1E−20 | 0.92 | 0.10 | 1.2E−14 | 8 | 103 | 1.07% | Both |
| X - 24455 | AFMID | 1.3E−12 | 0.71 | 0.10 | 1.1E−09 | 8 | 102 | 1.08% | Both |
| 3-Hydroxyisobutyrate | ALDH6A1 | 1.0E−09 | 0.66 | 0.11 | 3.2E−06 | 4 | 82 | 0.85% | B |
| X - 24807 | ALDH9A1 | 3.3E−15 | 0.44 | 0.06 | 7.5E−29 | 12 | 320 | 3.40% | Both |
| Phosphoethanolamine | ALPL | 1.4E−11 | 1.12 | 0.17 | 4.1E−11 | 3 | 33 | 0.35% | Both |
| Argininosuccinate | ASL | 4.8E−19 | 1.61 | 0.18 | 3.7E−16 | 6 | 29 | 0.30% | Both |
| X - 23654 | BBOX1 | 1.3E−06 | 0.57 | 0.12 | 1.7E−14 | 7 | 71 | 0.73% | S |
| X - 24801 | BBOX1 | 3.7E−08 | 0.64 | 0.12 | 2.1E−14 | 7 | 71 | 0.73% | S |
| Acisoga | CALY | 2.1E−06 | 0.45 | 0.10 | 1.5E−10 | 4 | 103 | 1.07% | S |
| Dimethylglycine | DMGDH | 2.4E−36 | 1.09 | 0.09 | 4.5E−34 | 10 | 127 | 1.31% | Both |
| N-acetyl-aspartyl-glutamate (NAAG) | FOLH1 | 2.1E−09 | 0.57 | 0.10 | 2.8E−10 | 4 | 96 | 0.99% | S |
| Trans-urocanate | HAL | 1.5E−11 | −0.47 | 0.07 | 3.0E−06 | 25 | 188 | 1.93% | B |
| X - 12410 | NAT1 | 1.0E−14 | −0.66 | 0.08 | 9.6E−13 | 4 | 129 | 1.32% | Both |
| 6-oxopiperidine-2-carboxylate | OPLAH | 1.3E−16 | 0.63 | 0.08 | 3.2E−14 | 15 | 164 | 1.97% | Both |
| N-acetyl-aspartyl-glutamate (NAAG) | OR5R1 | 1.6E−20 | −0.86 | 0.09 | 1.1E−21 | 8 | 79 | 0.81% | Both |
| 2'-O-methylcytidine | PHYHD1 | 5.4E−31 | 1.34 | 0.12 | 1.4E−24 | 11 | 60 | 0.62% | Both |
| 2'-O-methyluridine | PHYHD1 | 3.7E−26 | 1.43 | 0.14 | 7.7E−20 | 11 | 53 | 0.69% | Both |
| N-acetyl-beta-alanine | PTER | 3.4E−35 | 1.40 | 0.11 | 4.2E−29 | 6 | 72 | 0.75% | Both |
| N-acetyltaurine | PTER | 2.0E−26 | 1.24 | 0.12 | 7.6E−25 | 6 | 72 | 0.74% | Both |
| N6,N6-dimethyllysine | PYROXD2 | 2.1E−06 | −0.43 | 0.09 | 1.2E−13 | 10 | 120 | 1.23% | S |
| X - 21792 | RIOX1 | 7.0E−03 | −0.20 | 0.07 | 1.9E−11 | 8 | 191 | 1.98% | S |
| X - 10457 | RNPEP | 7.4E−14 | 1.06 | 0.14 | 3.0E−07 | 8 | 49 | 0.51% | B |
| Asparagine | SLC6A19 | 1.3E−07 | 0.70 | 0.13 | 1.1E−09 | 15 | 53 | 0.54% | S |
| Histidine | SLC6A19 | 1.1E−09 | 0.77 | 0.13 | 5.6E−11 | 15 | 53 | 0.55% | Both |
| Adipate (C6-DC) | SUGCT | 4.8E−12 | 0.63 | 0.09 | 1.2E−14 | 7 | 120 | 1.24% | Both |
| X - 22162 | TTC38 | 1.9E−22 | 0.76 | 0.08 | 9.5E−17 | 10 | 164 | 1.69% | Both |
| 3-Ureidoisobutyrate | UPB1 | 6.1E−25 | 1.37 | 0.13 | 1.3E−26 | 5 | 55 | 0.66% | Both |
| 3-Ureidopropionate | UPB1 | 7.3E−39 | 1.58 | 0.12 | 3.1E−44 | 6 | 64 | 0.66% | Both |

"Effect" denotes the cumulative impact of each copy of a rare, potentially damaging variant on metabolite levels, i.e. the average effect for each copy of a rare variant carried. Gene symbols are italicized. The same significance threshold for the burden and SKAT tests was chosen (two-sided test).
SE standard error, MAC minor allele count, MAF minor allele frequency.
[a]Significant in gene-based test: "B" means the association is significant in the burden test only, "S" means the association is significant in the SKAT test only, and "Both" means the association is significant in both tests.
[b]Biochemical name: the standard for this metabolite has not been run, but Metabolon, Inc. is highly confident in its identity.

**Table 2 Genes significantly associated with urine levels of ratios between fatty acids and amino acids.**

| Ratio of metabolites A / B | Gene | Burden | | | SKAT | | | No. of used SNPs | Total MAC |
|---|---|---|---|---|---|---|---|---|---|
| | | p-value | p-gain | Lower metabolite p-value | p-value | p-gain | Lower metabolite p-value | | |
| Trans-2-hexenoylglycine/hexanoylglycine | ACADM | 7.2E−33 | 1.4E−20 | 1.0E−12 B | 7.5E−42 | 1.3E+27 | 9.4E−15 B | 6 | 109 |
| 4-Methylhexanoylglutamine/hexanoylglutamine | ACY1 | 2.9E−13 | 3.0E+08 | 8.6E−05 B | 3.6E−13 | 3.8E+08 | 1.4E−04 B | 8 | 127 |
| N-acetylglycine/glycine | ACY1 | 1.7E−25 | 1.4E−06 | 2.4E−19 A | 1.4E−28 | 2.3E+08 | 3.1E−20 A | 8 | 123 |
| Cystathionine/1-carboxyethylisoleucine | CTH | 7.4E−24 | 4.6E+22 | 3.4E−01 A | 2.5E−24 | 3.2E+22 | 7.8E−02 A | 3 | 93 |
| Betaine/dimethylglycine | DMGDH | 1.9E−46 | 1.2E+10 | 2.4E−36 B | 1.4E−48 | 3.2E+14 | 4.5E−34 B | 10 | 128 |
| Tigloylglycine/isovalerylglycine | IVD | 2.2E−09 | 1.1E+05 | 2.3E−04 B | 4.3E−12 | 1.2E+07 | 5.1E−05 B | 3 | 18 |
| Tyrosine/phenylalanine | PAH | 4.2E−27 | 1.3E+22 | 5.4E−05 B | 5.4E−18 | 8.8E+14 | 4.7E−03 B | 12 | 85 |
| Lysine/threonine | SLC7A9 | 8.0E−12 | 2.7E+06 | 2.2E−05 A | 5.6E−10 | 2.3E+06 | 1.3E−03 A | 10 | 53 |

Gene symbols are shown italicized. Lower metabolite p-value: displays the lower of the two p-values from gene-based tests of the individual metabolites A and B that are part of the respective ratio. For each significant gene, only the ratio with the smallest p-value and largest p-gain is shown. The genome-wide significance threshold was p < 4.02E−11, and associations were filtered for p-gain >537,140 (Methods). The same significance threshold for the burden and SKAT tests was chosen (two-sided test).

metabolites are largely independent of common variants in the same region. Conversely, the rare variant association signals were abolished for *PYROXD2* and *RIOX1* and are therefore not independent of common genetic variants in the region.

**Properties of qualifying, metabolite-associated variants**. We next inspected the contributions of the individual qualifying variants to each of the 30 unique genes' aggregate association signals (Supplementary Data 3 for individual metabolites and Supplementary Data 6 for ratios). Because of the low number of carriers, most variants are not expected to show significant association by themselves, but still contain information on their effect direction. In general, largest effect sizes of qualifying variants on metabolites were observed for variants with lower minor allele frequency, and stop gain and splice alleles tended to have larger effects than nonsynonymous variants (Fig. 2). Across the 30 genes, 2272 out of a maximum of 4864 studied individuals (47%) carried at least one qualifying variant. Thus, although individually rare, at least one allele of a qualifying variant in the 30 genes was found in almost half of the study population, illustrating the high cumulative prevalence of their carrier status. There were genes in which all variants showed an effect in the same direction such as *RNPEP* and higher levels of the unnamed metabolite X–10457 (Supplementary Data 3), consistent with its detection by the burden test ($p = 7.4e−14$). *RNPEP* encodes aminopeptidase B, an exopeptidase responsible for the selective removal of arginine or lysine residues from certain peptides. Based on its mass and retention time (Supplementary Data 1), we hypothesize that the unknown metabolite X-10457 may represent an oligopeptide substrate of aminopeptidase B, whose concentrations increase upon loss of the enzyme's function. Potential candidates are listed in Supplementary Data 7.

Boxplots showing the distribution of analyzed metabolite levels by variant carrier status are shown in Supplementary Fig. 1 for each of the 53 significant gene-metabolite pairs. We distinguished qualifying variant carriers into heterozygous carriers of one variant per gene, heterozygous carriers of more than one variant per gene, and carriers homozygous for a rare variant. An example is shown in Fig. 3a, displaying argininosuccinate levels among individuals carrying and not carrying heterozygous variants in *ASL*. Variants included in gene-based association tests are aggregated based on their allele frequency and predicted functional effects. Figure 3a illustrates, however, that some of the aggregated variants based on these properties are likely to be neutral: for example, five of the six carriers of the missense variant encoded by rs143793815 (ASL p.T131M, NP_000039.2) had argininosuccinate levels below the median, while most carriers of the other aggregated variants showed clearly elevated levels well beyond the 75th percentile among controls. This result underscores that gene-based tests are powerful to detect metabolism-associated genes, but that individual variants still require experimental validation to establish their influence and magnitude of effect. Figure 3b illustrates the value of ratio analysis using *PAH* as an example.

**Connection of metabolite-associated genes to IEMs**. Some of the included rare variants are already known to cause IEMs when present in the homozygous state. An example is the adipate-associated *SUGCT* stop-gain variant p.Arg108Ter in the encoded succinyl-CoA:glutarate-CoA transferase (NM_024728.2), a known cause of autosomal-recessive glutaric aciduria III[30]. This enzyme catalyzes the succinyl-CoA-dependent conversion of glutarate to glutaryl-CoA, with adipate serving as an alternative Co-A acceptor to glutarate[31]. Loss of function consequently leads to higher levels of the acceptor molecules, such as glutarate and

**Table 3 Gene-based tests for significant metabolites before (left side) and after (right side) conditioning on common metabolite-associated variants in *cis*.**

| Gene | Metabolite | BURDEN | | | SKAT | top SNP in mGWAS | BURDEN | | | SKAT | Proportion of unconditional effect(%) |
|---|---|---|---|---|---|---|---|---|---|---|---|
| | | *p*-value | Effect | SE | *p*-value | | *p*-value | Effect | SE | *p*-value | |
| *ACADM* | Heptanoylglutamine | 1.2E−11 | 0.67 | 0.10 | 1.8E−13 | rs7513363 | 2.8E−09 | 0.58 | 0.10 | 2.3E−11 | 86% |
| | Hexanoylglutamine | 8.5E−13 | 0.69 | 0.10 | 2.0E−14 | | 6.1E−10 | 0.58 | 0.09 | 7.8E−12 | 85% |
| | Hexanoylglycine | 1.0E−12 | 0.66 | 0.09 | 9.4E−15 | | 6.1E−10 | 0.56 | 0.09 | 2.9E−12 | 85% |
| | Isocaproylglutamine | 1.3E−14 | 0.75 | 0.10 | 6.7E−16 | | 2.9E−11 | 0.63 | 0.09 | 5.4E−13 | 84% |
| | Isocaproylglycine | 4.3E−12 | 0.64 | 0.09 | 7.6E−13 | | 2.9E−09 | 0.53 | 0.09 | 2.0E−10 | 84% |
| | N-octanoylglutamine | 1.6E−15 | 0.77 | 0.10 | 1.1E−15 | | 8.3E−13 | 0.68 | 0.10 | 2.8E−13 | 89% |
| *AFMID* | N-formylanthranilic acid | 1.1E−20 | 0.92 | 0.10 | 1.2E−14 | rs72897838 | 4.3E−25 | 0.98 | 0.09 | 1.3E−17 | 107% |
| | X - 24455 | 1.3E−12 | 0.71 | 0.10 | 1.1E−09 | | 2.5E−14 | 0.75 | 0.10 | 6.1E−11 | 106% |
| *ALPL* | Phosphoethanolamine | 1.4E−11 | 1.12 | 0.17 | 4.1E−11 | rs1772719 | 1.1E−15 | 1.35 | 0.17 | 4.8E−15 | 120% |
| *FOLH1* | N-acetyl-aspartyl-glutamate (NAAG) | 2.1E−09 | 0.57 | 0.10 | 2.8E−10 | rs55728336 | 7.3E−09 | 0.50 | 0.09 | 2.2E−11 | 87% |
| *HAL* | Trans-urocanate | 1.5E−11 | −0.47 | 0.07 | 3.0E−06 | rs3213737 | 9.5E−11 | −0.44 | 0.07 | 5.4E−05 | 94% |
| *NAT1* | X - 12410 | 1.0E−14 | −0.66 | 0.08 | 9.6E−13 | rs35246381 | 9.2E−14 | −0.62 | 0.08 | 1.2E−11 | 95% |
| *PHYHD1* | 2'-O-methylcytidine | 5.4E−31 | 1.34 | 0.12 | 1.4E−24 | rs55758160 | 1.1E−34 | 1.33 | 0.11 | 1.7E−30 | 99% |
| *PYROXD2* | N6,N6-dimethyllysine | 2.1E−06 | −0.43 | 0.09 | 1.2E−13 | rs2147896 | 3.7E−01 | 0.06 | 0.06 | 2.1E−02 | <1% |
| *RIOX1* | X - 21792 | 7.0E−03 | −0.20 | 0.07 | 1.9E−11 | rs11626972 | 9.6E−01 | 0.00 | 0.06 | 2.1E−02 | 1% |
| *RNPEP* | X - 10457 | 7.4E−14 | 1.06 | 0.14 | 3.0E−07 | rs56768485 | 3.2E−13 | 0.97 | 0.13 | 2.5E−07 | 91% |
| *SLC7A9* | Lysine/threonine | 8.0E−12 | 0.89 | 0.13 | 5.6E−10 | rs12460876 | 8.3E−09 | 0.71 | 0.12 | 1.2E−06 | 80% |
| *TTC38* | X - 22162 | 1.9E−22 | 0.76 | 0.08 | 9.5E−17 | rs60032274 | 1.8E−23 | 0.75 | 0.07 | 4.2E−18 | 99% |

Gene symbols are shown italicized. The same significance threshold for the burden and SKAT tests was chosen (two-sided test).
mGWAS GWAS of the respective metabolites, SE standard error.

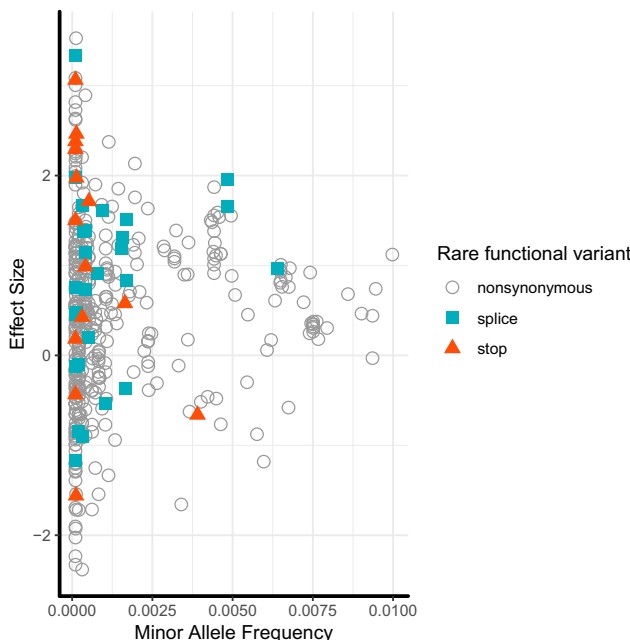

**Fig. 2 Effect size on metabolite levels (Y-axis) versus frequency (X-axis) of rare, putative damaging variants included in gene-based tests.** The symbols depict the effect of each variant that contributed to an aggregate association signal on the level of each significantly associated metabolite. Variants associated with more than one metabolite are therefore included for each metabolite because their associated effects can differ. The *Y*-axis represents the variants' effects on inverse normal transformed metabolite levels; one unit represents one standard deviation. Color-coding and marker shape represent the consequence of 218 variants in the 26 metabolite-associated genes: white circle (nonsynonymous), cyan square (splice), orange triangle (stop-gain/loss).

adipate, the readout in our study. Annotation of the 30 genes identified in our study showed that in fact 16 of them (53%) are known to underlie recessively inherited IEMs (Fig. 4, Supplementary Data 8 and 9). Whenever changes in metabolites were reported from patients with the respective IEM, the direction matched the one identified in our study (Supplementary Data 9). The remaining 14 genes may contain candidates for yet unreported human IEMs that could be detected from sequencing of patients presenting with extreme values of the metabolites implicated in our study. A literature search of these 14 genes confirmed that many connections with the implicated metabolites are supported by known biochemistry, such as *NAT1* and an N-acetylated metabolite. In the case of the *CALY* gene, however, the observed associations with polyamines may better be explained by nearby *PAOX* based on existing biochemical knowledge. Two of the 14 genes not yet implicated in IEMs have support from an earlier study of common variant effects on the plasma metabolome[32], and five additional gene-metabolite pairs for the 16 genes already known to underlie IEMs are supported by previous rare variant association studies of the plasma or serum metabolome (Supplementary Data 9).

**Whole-exome sequencing data from the UK Biobank allows for linking metabolite-associated genes to diseases.** We next investigated whether the burden of mostly heterozygous rare variants in the 30 identified genes was associated with human diseases beyond IEMs, capitalizing on the availability of whole-exome sequencing (WES) data from 50,000 white British participants of the UK Biobank Study[33]. We first evaluated a positive control, the association of *ALPL* variants (Supplementary Data 3) and blood levels of alkaline phosphatase, its gene product (Methods). The highly significant association (p-SKAT = 1e−24, p-burden = 1e−29) supported the use of UK Biobank data, and we subsequently investigated the aggregate effect of rare variants in the UK Biobank on 791 binary disease outcomes (Methods).

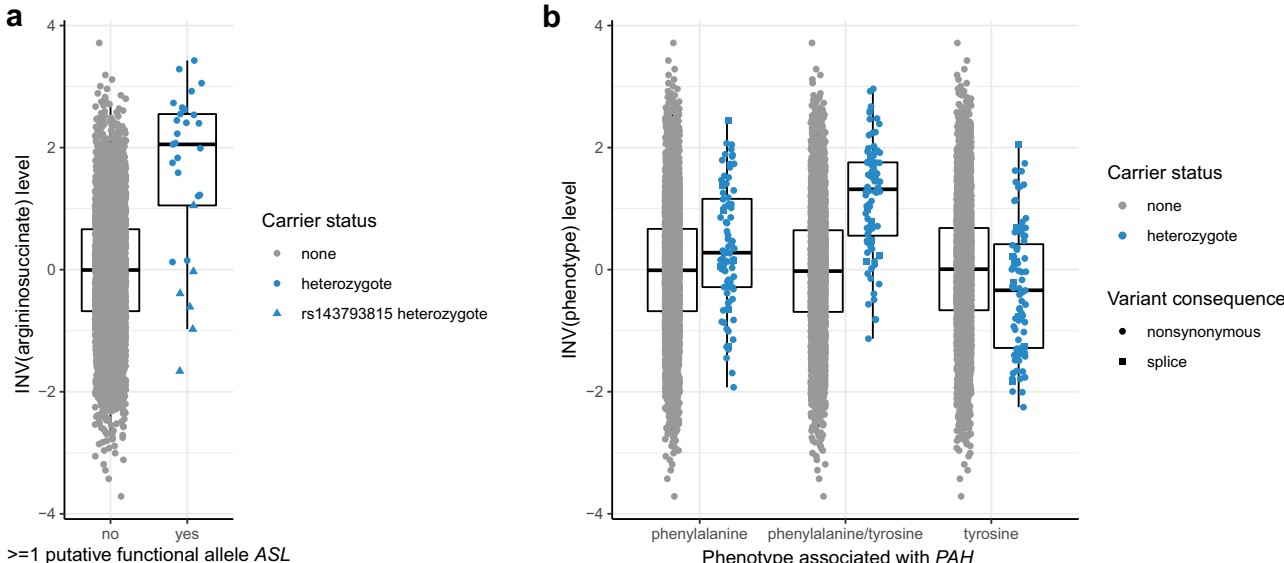

**Fig. 3 Effect size on metabolite levels (Y-axis) versus rare variant carrier status (X-axis).** Argininosuccinate levels are displayed by *ASL* rare variant carrier status (**a**) and levels of phenylalanine, tyrosine and their ratio for *PAH* rare variant carrier status (**b**). The Y axis represents metabolite levels after inverse normal transformation (INV), which allows for comparisons across metabolites. Units correspond to standard deviations. The symbol color indicates observed rare variant carrier status, and symbol shape variant consequence. The box ranges from the 25th to the 75th percentile of transformed metabolite levels, the median is indicated by a line, and whiskers end at the last observed value within 1.5*(interquartile range) away from the box. Sample sizes: $n = 4863$ individuals for argininosuccinate; $n = 4839$ individuals for phenylalanine, phenylalanine/tyrosine ratio and tyrosine.

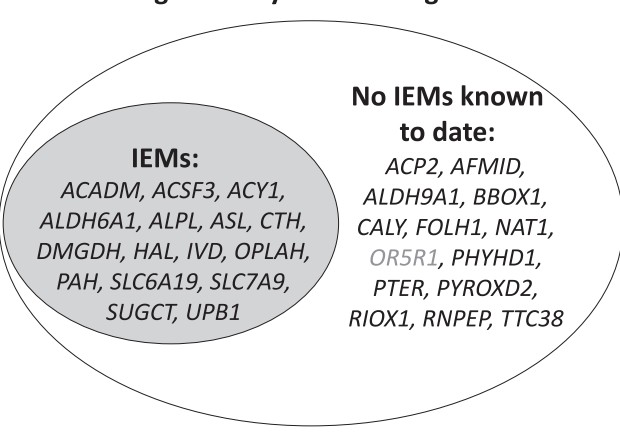

**Fig. 4 Venn diagram of the involvement of identified genes in inborn errors of metabolism (IEMs).** All of the known IEMs are recessively inherited. Genes for which no IEMs are yet identified represent candidate genes in individuals with extreme levels of the implicated metabolites. Gray font is used for *OR5R1* because of the extended linkage disequilibrium in the region extending to *FOLH1*, which may therefore not represent an independent signal.

While there were no associations that reached statistical significance after multiple testing correction ($p < 6.3e{-}05$; 0.05/791), three associations showed suggestive evidence of association (Supplementary Data 10). The association between loss-of-function alleles in alkaline phosphatase and fractures of the pelvis ($p = 9.6e{-}06$) is consistent with the enzyme's role in systemic phosphate homeostasis, and with defective bone mineralization in the corresponding IEM hypophosphatasia (MIM #241500). The association between rare variants in *ALDH9A1* and hypotension has not previously been reported and is interesting insofar as the inhibition of another aldehyde dehydrogenase, ALDH2, has been linked to hypotension in response to ethanol in rats[34], and the

locus is detected in genome-wide association studies for blood pressure – alcohol interaction in humans[35]. Together, these findings illustrate the potential of linking metabolite-associated genes with health outcomes in large biobank studies with medical record linkage to more broadly understand the health consequences of inter-individual differences in metabolism.

**Enrichment analyses nominate target tissues and cell types in kidney and liver as well as implicated terms and pathways.** Given the quantification of metabolites from urine and the central role of the kidney in determining final urine metabolite concentrations, we investigated in which tissues and kidney cell types the 30 significant genes were highly expressed using human gene expression data from the GTEx Project[36] as well as bulk and single-cell and single-nucleus RNA-sequencing data from human, rat, and mouse kidney[37–40]. Figure 5a shows tissue-specific high expression of 29 available genes across 39 non-brain tissues. A strong signature of high expression in liver for many of the genes suggests that the urinary concentration of their associated metabolites may be mainly determined by their generation followed by free glomerular filtration and/or tubular secretion. There were also genes with high expression in the small intestine, such as *SLC6A19* and *SLC7A9*. Several genes also showed high expression in the kidney cortex in comparison to other tissues (Methods).

Since the number of kidney samples even in the final release of the GTEx data is relatively small, and given the high importance of the kidney in determining urine metabolite concentrations, we next investigated several kidney-specific datasets. Across 14 cell types from micro-dissected rat kidney, we found that many of the 23 genes available for evaluation were highly expressed in the three segments of the proximal tubule both at the level of the transcriptome[40] and of the proteome[39] (Fig. 5b). Epithelial cells of the proximal tubule have high metabolic activity and are responsible for the bulk reabsorption of many filtered solutes and small molecules such as metabolites. The cell-type-specific expression matched prior knowledge from experimental studies

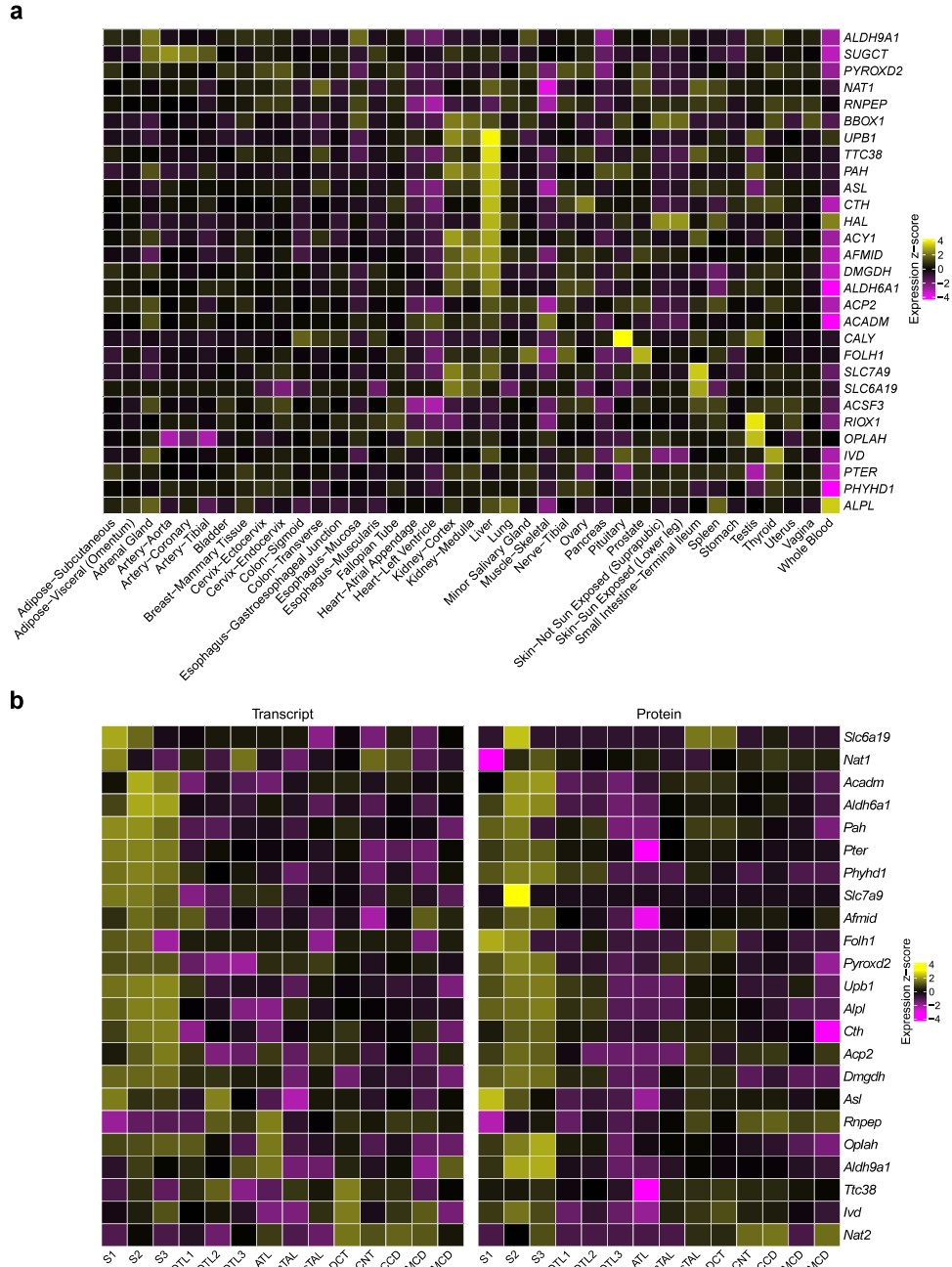

**Fig. 5 Enrichment analyses highlight specific tissues and cell types in which the identified genes are highly expressed.** Human tissues (**a**) as well as rat micro-dissected kidney tubule cell types (**b**) were evaluated. Tissues are based on GTEx Project data V8, transcriptome data of kidney tubule cell types on the publication by Lee et al.[40] and proteome data of same cell types on the publication by Limbutara et al.[39]. **a** Gene expression levels are from 39 non-brain tissues from GTEx V8. Presented values are the mean of $\log_{10}$-transformed Transcript Per Million (TPM) of samples in each tissue. Only 29 genes are displayed because *OR5R1* is not included in GTEx V8. **b** S1, first segment of proximal tubule; S2, second segment of proximal tubule; S3, third segment of proximal tubule; DTL1, descending thin limb type 1 (short-looped nephron); DTL2, descending thin limb type 2 (long-looped nephron); DTL3, descending thin limb type 3 (long-looped nephron); ATL, ascending thin limb; mTAL, medullary thick ascending limb; cTAL, cortical thick ascending limb; DCT, distal convoluted tubule; CNT, connecting tubule; CCD, cortical collecting duct; OMCD, outer medullary collecting duct; IMCD, inner medullary collecting duct. Presented values are the mean of $\log_{10}$-transformed TPM of samples in each cell type. There was no rat homolog for *SUGCT*; *NAT1* has two homologues (*NAT1* and *NAT2*). *ACSF3*, *ACY1*, *BBOX1*, *CALY*, *HAL*, *OR5R1*, and *RIOX1* were not included in the transcriptome dataset. Source data are provided as a Source Data file.

for some of the genes and can therefore be considered positive controls, such as expression of SLC6A19 in proximal tubular cells detected by immunohistochemistry[41]. Of note, many genes that did not show high kidney expression in comparison to extra-renal tissues showed high expression in specific kidney cell types. Single-cell gene expression data from humans and mice

confirmed a signature of high expression in renal proximal tubule cells, indicating conserved functions throughout evolution (Supplementary Figs. 2 and 3). This suggests that cell type heterogeneity in tissue from solid organs may mask cell-type-specific signatures. We therefore tested enrichment for high expression of the 30 genes both across and within tissues

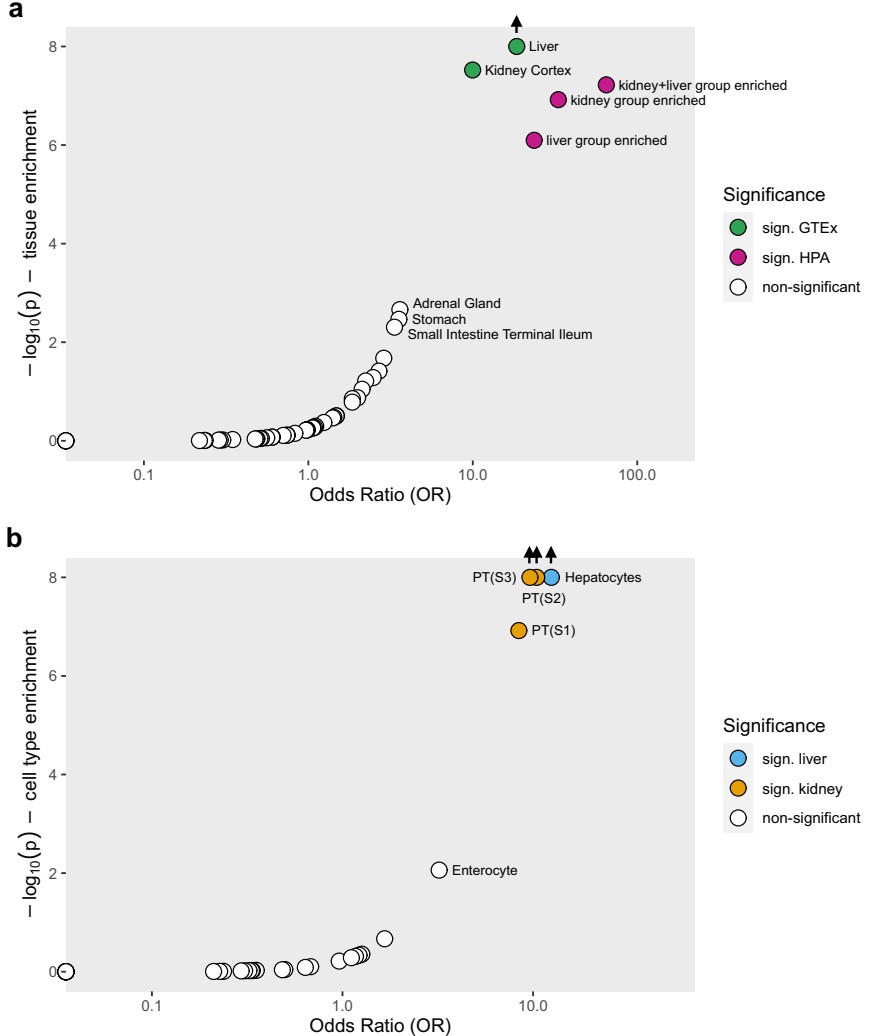

**Fig. 6 Enrichment analysis of the 30 significantly associated genes in human tissues and cell types.** Enrichment analyses were performed across 39 human tissues represented in GTEx Project V8, and in liver, kidney and both liver and kidney from the Human Protein Atlas (HPA) (**a**); and across cell types from kidney, liver and gut based on single-cell RNA-sequencing data (**b**). Source data are provided as a Source Data file.

(Methods). Across tissues, we found significant enrichment in liver (odds ratio [OR] = 18.5) and kidney (OR = 11.6) (Fig. 6). Within tissues, enrichment was identified in hepatocytes in the liver (OR = 12.5) and in proximal tubule cells in several scRNA-seq datasets from the kidney (OR range 8.4–29.6). Since kidney and liver are known to share important transcriptional programs[42], it may however be more interesting to test for enrichment of co-expressed genes rather than of genes that show high enrichment in either liver or kidney cell types. Indeed, the 30 genes showed the greatest enrichment when testing for enrichment among all genes highly expressed in both liver and kidney (OR = 64.9, Fig. 6, Methods), underscoring the coordinated role of these two organs in human metabolism.

While associations with individual metabolites and metabolite ratios were largely biologically plausible and agreed with clinical findings from patients with the corresponding IEMs, they do not answer questions about potential shared pathways. We thus performed enrichment testing of the 30 genes based on terms in the Gene Ontology (GO) and the KEGG database (Methods). We identified 33 terms significantly enriched for the identified genes (Supplementary Data 11), with the greatest enrichment (OR = 386) observed for the GO biological process "amino-acid betaine

biosynthetic process", with "amino-acid betaine metabolic process" and "carnitine metabolic process" being other highly enriched biological terms (both OR = 145). The greatest number of genes, 25 out of 30, mapped into the enriched molecular function pathway "catalytic activity", consistent with their functions as enzymes. The strong representation of terms, pathways, functions and processes related to amino-acid metabolism is in agreement with the important role of the kidney in determining their urine concentrations and with the modeling of amino acid ratios, facilitating the detection of enzymes and transporters for amino acid handling.

**Use of a virtual metabolic human to validate significant gene-metabolite associations.** Lastly, while genome- and exome-wide association studies are powerful tools to reveal statistical associations between metabolite levels and genetic variation, it can be challenging to derive mechanistic hypotheses underlying these associations. We therefore used constraint-based modeling and reconstruction analyses (COBRA) for mechanistic network modeling to generate in silico knockouts of the implicated genes in a whole-body, organ-resolved model of human metabolism integrating physiological traits[43] (Methods) based on the virtual

**Table 4 Comparison of in silico knockout whole-body metabolic models versus healthy whole-body metabolic models using COBRA modeling with GWAS results.**

| Gene | Metabolite | VMH ID | In silico maximal flux of urine metabolite excretion | Observed effect on urine metabolite concentrations | In silico maximal flux in the blood compartment |
|---|---|---|---|---|---|
| *HAL* | Trans-urocanate | urcan | ↓ | ↓ | ↓ |
| *ALPL* | Phosphoethanolamine | ethamp | = | ↑ | ↑ |
| *ASL*[a] | Argininosuccinate | argsuc | ↑ | ↑ | ↑ |
| *ALDH6A1* | 3-Hydroxyisobutyrate | 3hmp | ↑ | ↑ | ↑ |
| *DMGDH* | Dimethylglycine | dmgly | ↑ | ↑ | ↑ |
| *ACY1* | N-acetylglycine | acgly | ↑ | ↑ | ↑ |
| | N-acetylglutamate | acglu | ↑ | ↑ | ↑ |
| *UPB1* | 3-Ureidoisobutyrate | 3uib | ↑ | ↑ | ↑ |
| | 3-Ureidopropionate | cala | ↑ | ↑ | ↑ |

"↓" denotes decreased maximal flux in knockout model, "↑" denotes increased maximal flux in knockout model, "=" denotes unchanged flux in knockout model. VMH – virtual metabolic human database, www.vmh.life.
[a]This knockout model and the corresponding flux simulation were already reported in Thiele et al.[43]. Gene symbols are shown italicized.

metabolic human database[4]. Indeed, we found that association results from the GCKD study population could be successfully recapitulated for *DMGDH, ALPL, ALDH6A1, HAL, ACY1, UPB1,* and *ASL*, of which the associated metabolites were included into the whole-body metabolic reconstruction and hence modeling was possible. As shown in Table 4, the direction of effect on urine metabolite concentrations was correctly predicted by the in silico knockout models for eight of the nine metabolites, which was significantly better than chance (binomial test: $p = 8.9e-4$). An exception was phosphoethanolamine, for which the in silico knockout of *ALPL* in humans was predicted to result in unchanged levels of the metabolite in urine, but higher levels were observed. A potential explanation for this finding is that three tissue-specific isoforms of the encoded enzyme are known to exist in the liver, kidney, and bone. Urine concentrations may be influenced especially by the kidney-specific isoform that may differ in non-modeled properties from other isoforms of the enzyme. In any case, the integration of genetic association studies in populations with mechanistic network modeling in the environment of genome-scale whole-body models supports that the rare variants in the modeled genes that were associated with the corresponding metabolites in the GCKD study are indeed loss of function mutations. The predicted direction of effects on blood metabolite concentrations (Table 4) are consistent with reports of the observed changes among patients affected by the corresponding IEM in all instances. Together, these findings can be regarded as proof of principle that COBRA modeling can be applied as a further layer of validation for results from association studies in human population studies, specifically when experimental validation may not be possible or ethical. Note that the utilized whole body model was never trained to predict metabolite rare variant associations or the effects of IEMs.

## Discussion
In this study of genetic determinants of the urine concentrations of 1487 metabolites and 53,714 metabolite ratios, we identified 128 significant associations between levels of metabolites or metabolite ratios and the cumulative effect of rare, potentially functional variants in 30 genes that were almost exclusively present in the heterozygous state and cumulatively identified in 47% of studied individuals. The fact that more than half of the implicated genes have been identified as causes of recessively inherited IEMs underscores the potential of our approach to reveal variants and genes not yet known to cause metabolic diseases. Our well-powered study of the aggregate effect of rare variants on urine metabolite concentrations reveals genes known

to affect their concentrations in blood, but also detects effects on urine metabolite concentrations likely to represent their kidney-specific handling and only detectable when studying urine. In silico validation of knockouts of implicated genes in a virtual metabolic human correctly predicted the direction of observed changes in metabolite levels.

Previous studies reporting on the association of rare genetic variants and metabolite concentrations have mainly quantified metabolites from blood[18,19]. An exome array study of 217 plasma metabolites from 2,076 population-based individuals identified associations between rare variants in *HAL* and histidine, *PAH* and phenylalanine, and *UPB1* and ureidopropionate[18]. Our study extends the latter finding in the same direction to urine. Rare variants in *HAL* were associated with trans-urocanate rather than histidine levels in our study. The encoded histidase catalyzes the deamination of L-histidine to trans-urocanic acid[44]. Histidine was quantified in our study but not associated with *HAL*. A potential explanation for this finding is the differential handling of histidine and trans-urocanate along the nephron, which contains specific transport proteins to reabsorb essential amino acids such as histidine, thereby altering their urine concentrations after filtration from blood. The association of urine histidine levels with rare variants in *SLC6A19* in our study highlights the encoded amino acid transporter as the relevant transport protein. This is in agreement with findings from patients with Hartnup disease, where bi-allelic loss-of-function mutations in *SLC6A19* lead to a reabsorption defect of histidine in renal tubular cells[45]. Associations between rare variants and lipid metabolites on the other hand may be better detected in blood, such as for *APOA5* and diacylglycerol[18], consistent with extra-renal handling and/or excretion. The study of different biospecimens therefore provides complementary insights into different aspects of human metabolism. While we found that seven gene-metabolite associations identified in our study of urine specimens matched earlier findings from studies of serum or plasma, a systematic assessment of overlap is complicated because of differences in metabolite platforms, genotyping and sample size.

The only previous study investigating the effects of common and rare genetic variants on urine metabolite concentrations was based on 193 individuals who developed CKD and 193 controls[22]. This previous study quantified only 154 metabolites using a different mass spectrometry-based platform, as compared to our study of 4864 persons and 1487 metabolites and their ratios. While the previous study reported some associations between individual rare variants and urine metabolites, it was underpowered to detect any significant associations at the gene-level.

Thus, the identified 128 associations between the aggregate effects of rare genetic variants and the concentrations of urine metabolites have not been reported previously.

The great majority of individuals examined in this study had reduced eGFR, indicating impaired renal filtration function[46], raising the question whether the findings likely also apply to population-based samples. Indeed, we recently showed very good agreement of genetic effects on urine metabolite concentrations between the GCKD patients and individuals from the general population[17]. In addition, genes detected in this study were also identified in association studies of blood metabolite concentrations in population-based samples[19]. Lastly, 16 of the 30 genes identified in this study are known to cause recessively inherited IEMs with corresponding metabolic abnormalities, further supporting that findings from this study extend beyond individuals with reduced eGFR. The study of individuals with impaired kidney function, however, may additionally allow for the detection of genes linked to the metabolism of uremic toxins. The suggestive associations of several genes with different acetylspermidine metabolites in our study may suggest such a finding, but requires confirmation and testing in population-based samples that are currently unavailable.

There are several insights of potential clinical relevance. First, several of the rare variants identified in the heterozygous state in our study are known causes of IEMs when present in the homozygous state, such as the adipate-associated *SUGCT* stop-gain variant encoding p.Arg108Ter (NM_024728.2), a known cause of autosomal-recessive glutaric aciduria III[30]. In general, heterozygous carriers of causative mutations for IEMs are not clinically affected, suggesting that the residual function of the encoded proteins is sufficient to perform their physiological functions. However, the resulting changes in metabolite levels were still sufficient to identify the respective genes in our study. Thus, genes identified in our study that are not yet known to cause IEMs represent candidates for targeted assessment in patients with a suspected IEM who feature abnormal levels of the implicated metabolite. Second, genes identified in this screen may also contain common variants associated in GWAS with more common diseases, in which the implicated metabolite(s) may be involved. This is exemplified by a reported association of common variants in the *SUGCT* locus in a GWAS of migraine[47], where identification of the causal gene within the locus is a classical challenge. Our study directly connects *SUGCT* to glutarate and adipate levels and to monogenic glutaric aciduria III. A previous report of a patient with glutaric aciduria III presenting with migraine[48] now supports the hypothesis that *SUGCT* may be the underlying causal gene for the GWAS signal, with a potential role as a mediator or biomarker of migraine that deserves further study. Along the same lines, our study implicates glycochenodeoxycholate 3-sulfate-associated *SLC10A2* as the causal gene in a locus identified in GWAS of gallstone disease[49] by detecting rare, metabolite-associated variants in this gene.

As a conceptual add-on, we successfully integrated our results with COBRA modeling and demonstrated that rare variant association results and in silico knockout models of human metabolism showed very close agreement. This opens up new research routes, as methods based on mechanistic modeling, in the case of COBRA via the analyses of steady state solution spaces of biochemical reaction networks, are well suited for individualized phenotype predictions. This is because COBRA modeling does not rely on population statistical frameworks, such as association studies or machine learning, and is therefore applicable when small sample size complicates statistical inferences, while COBRA models still allow personalized parametrization of models via Omics measurements[43]. However, COBRA models are still missing metabolites or genes, and are continuously updated and expanded by the systems biology community, as it is the case for human[50–52], yeast[53,54], and well-studied microbes such as *Escherichia coli*[55,56]. Missing information may limit their applicability; the integration of genetic association results helps with identifying such missing information and can lead to their refinement. This is exemplified in our study, where the observed genetic association indicated that the COBRA model did not produce correct results in the case of *ALPL* knockout. A potential explanation for this finding is the lack of incorporation of the known presence of organ-specific ALPL isozymes into the model. In conclusion, COBRA modeling and genetic association studies complement each other, and this study is a step towards bridging the worlds of population statistics and mechanistic modeling.

Some aspects of our study warrant mention: because of the array-based design, not all rare or private genetic variants of potentially large effect are likely to be captured, and not all genes are equally well represented. These limitations can be addressed by future whole-exome or whole-genome sequencing studies. The study population was restricted to individuals of European ancestry, and studies of persons of non-European ancestry are likely to reveal additional findings. The special study sample, with the quantification of almost 1500 metabolites from urine of almost 5,000 individuals, currently precludes external validation of detected associations. However, the biological plausibility, good agreement with findings from blood-based studies and with genes implicated in corresponding IEMs, as well as the conservative correction for multiple testing bolsters confidence that at least the great majority of reported associations represent true positive findings. Lastly, the identity of some of the associated metabolites is yet unknown. We believe that unsupervised clustering of known and not yet identified molecules, combined with knowledge about the unknown molecule's mass spectrometry information and the implicated gene's function, offers a promising way forward to reveal their identity. In fact, we previously used this approach to reduce the search space and were able to successfully de-orphanize a formerly unidentified molecule[17].

In summary, our study extends the map of genes influencing urine metabolite concentrations, illuminates ongoing metabolic processes at the systemic and organ level, implicates candidate variants for known IEMs and genes for yet unknown IEMs, and highlights the potential of linking population genetics to whole-body, organ-resolved models of human metabolism.

## Methods

**Study design and participants**. The German Chronic Kidney Disease (GCKD) study is an ongoing multi-center prospective cohort study of patients with CKD under regular care of a nephrologist. It was registered in the national registry for clinical studies (DRKS 00003971) and approved by the local ethics committees of the participating institutions (Supplementary Note 1). Between 2010 and 2012, 5217 patients aged 18–74 years provided written informed consent and were enrolled into the study[57]. Eligible patients had a glomerular filtration rate estimated from serum creatinine (eGFR) between 30 and 60 ml/min/1.73 m$^2$, or an eGFR >60 ml/min/1.73 m$^2$ in combination with a urinary albumin-to-creatinine ratio (UACR) of ≥300 mg/g or albuminuria of ≥500 mg/d or the corresponding values of protein in urine[57]. Clinical data, medical and family history, medications, socio-demographic factors were collected by trained study personnel using standardized instruments. At the baseline visit, blood was drawn to obtain serum and plasma, and a spot urine sample was collected. Biomaterials were immediately processed following a standardized protocol and shipped frozen to and stored at −80 degrees Celsius in a central biobank for future analyses[58]. More detailed information about the design of the study and the recruited study population can be found in previous publications[46,57].

**Genotyping and imputation**. A detailed description of genotyping and the subsequent data cleaning in the GCKD study has been published previously[29]. In brief, genomic DNA was extracted from whole blood and was available for 5123 GCKD participants. Genotyping was carried out using Omni2.5Exome BeadChip arrays (Illumina, GenomeStudio, GenotypingModule Version 1.9.4). Quality control of genotypes was performed for individuals and SNPs for the ExomeChip content of the array, including checks specific to exome chip data as published previously[59].

Briefly, based on the quality control steps, 96 individuals and 3,818 SNPs were removed, the latter of which did not pass the filters of call rate >95% and Hardy–Weinberg equilibrium $P$-value $\geq 1E-5$. The cleaned Exome Chip dataset contained genotypes of 226,233 variants from 5,027 individuals and was post-processed by zCall[60] with a z-score threshold of six and then used in the Exome Chip association analyses.

**Metabolite measurements**. Metabolites were quantified from urine specimens collected from 5096 GCKD study participants at the baseline visit at Metabolon, Inc. (Research Triangle Park, NC) with ultra-high performance liquid chromatography-tandem mass spectrometry (UPLC-MS/MS) non-targeted methods including compound identification and relative quantification. Sample preparation and mass detection were carried out with the technical proposal published previously[61]. In short, the platforms used a Waters ACQUITY UPLC and a Thermo Scientific Q-Exactive high resolution/accurate mass spectrometer with an electrospray ionization (ESI) source and Orbitrap mass analyzer at 35,000 mass resolution. The UPLC methods were optimized for hydrophilic and hydrophobic molecules and the mass analyzer applied in positive and negative modes for a total of four UPLC-MS/MS methods per sample. The scan range was 70 to 1000 $m/z$.

Software developed by Metabolon facilitated the identification of metabolites based on a comparison of ion features in the GCKD samples to a reference library that contained entries on molecular weight ($m/z$), retention time, preferred adducts, in-source fragments and associated MS spectra of chemical standards. The area under the curve was used for peak quantification, and raw area counts were normalized, for each metabolite in each sample, in order to account for inter-day instrument variation. More details can be found in the Supplementary Note 2.

**Quality control and data cleaning of metabolomics data**. Quality control of the urine metabolite data in the GCKD study has been published in detail previously[17]. In brief, samples and metabolites were evaluated for missing and outlying values as well as metabolite variance. Outlying values (>5 SD) for each metabolite were set to missing. To account for inter-individual differences in urine dilution, metabolite concentrations were normalized using the probabilistic quotient[62], which was derived from endogenous metabolites with <1% missing values. In order to avoid false positive associations due to the small sample size, only metabolites or metabolite ratios quantified in at least 300 individuals were analyzed. Data were inverse normal transformed prior to analysis. After removal of samples without genotypes, metabolites, or covariates, the final dataset for the evaluation of 1487 individual metabolites (Supplementary Data 1) consisted of up to 4864 individuals. Pairwise metabolite ratios were computed across 265 amino acids, 32 peptides and 32 lipid-fatty acid metabolites, resulting in 53,714 ratios metabolite ratios with at least 300 measurements available in up to 4795 individuals.

**Additional variables**. Serum creatinine used for the estimation of GFR was measured by an IDMS traceable enzymatic assay (Creatinine plus, Roche). The four-variable Chronic Kidney Disease Epidemiology Collaboration (CKDEpi) equation was used to estimate GFR[63]. The UACR was calculated from urinary albumin and creatinine measurements, with creatinine measured using the same assay as in serum, and albumin with the ALBU-XS assay (Roche/Hitachi Diagnostics GmbH, Mannheim, Germany).

**Rare variants association studies based on exome chip**. Two types of rare variant aggregation tests were conducted, the burden test[64] and the sequence kernel association test (SKAT)[65] as implemented in the R package seqMeta (v1.6.7)[66] and adjusting for age, sex, eGFR, UACR, and the first three principal components. Potentially deleterious variants that were aggregated within each gene were defined as those with MAF < 1% and presumably having a major effect on the gene product (nonsynonymous, stop gain/loss, splicing; "qualifying variants") as annotated using dbNSFP v.2.0[67,68]. Results were filtered to retain genes with cumulative minor allele count $\geq 10$ and with $\geq 2$ contributing variants per gene. In the analysis of individual metabolites, up to 11,552 genes passed the filtering criteria. To account for multiple testing, the overall type I error of 0.05 was adjusted for the number of metabolites ($n = 1487$), the number of genes ($n = 11,552$), and the number of performed aggregation tests ($n = 2$) using a Bonferroni correction. Statistical significance was thus defined as $p < 1.46E-09$. In the analysis of the 53,714 metabolite ratios, up to 11,587 genes passed the filtering criteria; therefore, the corresponding significance threshold adjusted for multiple testing was set at $p < 4.02E-11$. Suggestive significance for both metabolites and their ratios was defined as $p < 2.2E-06$, corresponding to the testing of one metabolite. The association between metabolites and metabolite ratios and individual genetic variants that contributed to the significant associations identified in gene-based tests was evaluated using linear regression and an additive genetic model adjusting for the same covariables as implemented in the R package seqMeta (v1.6.7)[66]. The $p$-gain statistic as a measure of information gain of a metabolite ratio in comparison to its individual components was computed as the minimum of the two $p$-values of the individual metabolites divided by the $p$-value of the ratio[69]. Only metabolite ratios with a $p$-gain of at least 10 times the number of evaluated ratios[69], i.e. 537,140, were retained.

The potential influence of common variants on the detected gene-metabolite association signals was evaluated. For each significant metabolite or metabolite ratio identified in this study, we assessed whether there was a genome-wide significant common variant association signal in GWAS of the implicated metabolites, which we published previously[17]. When the corresponding gene implicated in this study mapped within ±500 kb of a GWAS index SNP for the respective metabolite, we repeated both rare variant aggregation analyses conditional on the genotype of the corresponding index SNP. Change in effect size from the burden test before and after conditional analysis was also assessed.

**Connection of implicated genes to health outcomes**. The aggregated effect of the rare variants on health outcomes were investigated in the UK Biobank WES dataset (application no. 28807)[70]. The analysis for the positive control gene, *ALPL*, was conducted in the European subpopulation ($N = 41,045$), excluding subjects with missing values, using the same rare variants, same transformation, and the burden and SKAT test as described before with a similar covariate set as in the GCKD study: age, sex, and the first three principal components. Second, the association between the aggregated effects of rare variants in the genes of interest were evaluated in the whole-exome sequencing dataset of the UK Biobank with 791 medical outcomes using gene-based linear mixed models as implemented in the SAIGE-GENE software and detailed in[71].

**Statistical power**. The statistical power to detect associations between metabolites and rare genetic variants in our study of 4864 unrelated individuals across a range of MAFs and effect sizes was calculated using Quanto (v1.2.4) and shown in Supplementary Data 12.

**Gene expression data sources and analyses**. GTEx RNA-seq V8 data[36] was downloaded from GTEx Portal Datasets page (Data Availability). The TPM matrix was filtered according to criteria used in Finucane et al.[72], resulting in expression levels of 33,507 genes for 17,382 samples spanning 54 tissues. RNA-seq as well as protein mass spectrometry data from micro-dissected rat tubules was downloaded from Kidney Tubules Expression Atlas data table page (Data Availability)[39,40]. Kidney single-cell RNA-seq data from human[38] and mouse[37] were downloaded from GEO (Data Availability), and processed in R package Seurat[73] based on the methods in the respective original publications[37,38]. The human kidney dataset contained 4524 cells in 17 cell clusters, and the mouse kidney dataset contained 43,745 cells in 16 clusters. The normalized count matrix from the published Human Liver Cell Atlas[74] included 10,372 cells and was also processed in Seurat. Cells in 39 clusters were merged into 10 major liver cell types. The intestine single-cell data from Wang et al.[75] was provided as a Seurat file by the authors. The dataset included 14,537 epithelial cells from human ileum, colon and rectum in 7 clusters. For each expression dataset, a list of specifically expressed genes was generated for each tissue or cell type by taking the top 10% of genes ranked by t-statistics extracted from linear models[72]. For visualization of the expression of metabolite-associated genes as a heatmap, the mean expression values in each tissue or cell type were z-score transformed.

**GO, KEGG, tissue and cell-type enrichment analyses**. We generated a database of Entrez gene identifiers using the Bioconductor R database org.Hs.eg.db v3.10.0 and KEGG.db v3.2.3 that contained, for each gene, Gene Ontology (GO) terms, Kyoto Encyclopedia of Genes and Genomes (KEGG) pathways, and the top 10% highly expressed genes in each GTEx V8 tissue, the kidney- and liver-enriched as well as group-enriched genes downloaded from the Human Protein Atlas (HPA, Data Availability) and human kidney, liver, and intestine as well as murine kidney cell types[37,38,74–76]. The database was filtered for terms with at least five genes. For enrichment testing, the observed number of the 30 identified genes in the GO terms, KEGG pathways, as well as in the top 10% highly expressed genes in each GTEx tissue and kidney, liver and intestine cell type was compared to the number obtained from lists of 30 randomly drawn genes (GO and KEGG: 1e7 draws; others: 1e8 draws). Multiple testing correction was performed using the Bonferroni method[77] for GTEx tissues (0.05/51) and subsequently tested organ-specific cell types, and the Benjamini-Hochberg procedure[78] for GO and KEGG terms (FDR < 0.05).

**Constraint-based modeling and reconstruction analyses (COBRA)**. Aiming at narrowing the gap between genetic association studies and mechanistic insights, we integrated findings from the exome-wide association studies of rare variants with the frameworks of COBRA, complimenting thereby statistical association analyses with mechanistic network modeling. In COBRA, an in silico model of an organism is assembled based on genomic, biochemical, and physiological data[79,80], basically deriving the space of possible thermodynamically feasible steady-state solutions for the organism's metabolism. The mathematical and computational framework allows then for investigating perturbations of the system in silico, such as gene knockout models. This computational modeling approach allows for in silico prediction of metabolic alterations in IEMs[51,81,82]. Here, we used a male whole-body, organ-resolved model of human metabolism (male WBM)[43], which consisted of 81,094 reactions, 56,452 metabolites, and 1681 genes across 22 organs, 6 blood cells, and 13 biofluid compartments, including urine. The WBM reconstruction has been built using the most recent version of the generic genome-scale reconstruction of human metabolism, Recon 3D[52]. The WBM model was constrained using the constraints for the reference

man, as described previously[43]. Briefly, the constraints included maximal excretion rates for the considered 920 urine metabolites, based on reported urine concentrations[83] and a urine production rate of 2 l/day. The diet was constrained to an average European diet. To model the gene knockouts, we first identified all reactions carried out by the corresponding encoded enzymes, using the virtual metabolic human database[4] (www.vmh.life). We only considered genes that were present in the WBM reconstruction and had no isozymes in the model and metabolites where urine excretion reactions were defined in the WBM reconstruction. To determine whether the urine excretion of the metabolite would be altered in silico in the knockout, we first set the flux bounds, or constraints, of the identified corresponding reaction(s) in all organs that contained the reaction(s) to zero. We then set the lower and upper bounds of urine excretion reactions to zero and infinity, respectively, for each metabolite found to be significantly associated with a gene in this study (e.g., EX_ dmgly[u] for dimethylglycine). We subsequently maximized the flux through the urine excretion reaction individually. The maximal possible flux value was recorded and compared to that of the healthy model. The healthy case was simulated by first maximizing the flux through all organ-specific reactions associated with one of the genes of interest, followed by setting this maximal flux value as constraint on these reactions. We maximized flux through each of the urine metabolite excretion reactions after setting the corresponding reaction constraints to zero and infinity, respectively, as described above. To simulate changes in the blood compartment, we added for each of the metabolites a so-called demand reaction, which allows for their accumulation in the blood compartment, in the form of, e.g., DM_ dmgly[bc] for dimethylglycine. We repeated the aforementioned simulations but this time we maximized the flux through each of the demand reactions for each corresponding healthy and knockout model. All simulations were carried out using Matlab v2018b (Mathworks, Inc.) as simulation environment and Ilog Cplex v10.12 (IBM, Inc.) as linear programming solver, the COBRA Toolbox v3.0[84], and the physiologically and stoichiometrically constrained modeling (PSCM) toolbox v1.0[43].

**Reporting summary**. Further information on research design is available in the Nature Research Reporting Summary linked to this article.

## Data availability

In accordance with the informed consent, individual-level data of the GCKD study can be shared with the scientific community for dedicated research projects upon application. The study website (https://www.gckd.de) contains information about the application and publication processes. mzML files representing examples of the unknown molecules in Table 1 are available through the EMBL-EBI database Metabolights under study ID MTBLS284. GTEx RNA-seq V8 data were downloaded from GTEx Portal Datasets page (https://www.gtexportal.org/home/datasets/). Count matrices of kidney single-cell RNA-seq data from human and mouse were downloaded from GEO under accession numbers GSE118184 and GSE107585. RNA-seq as well as protein mass spectrometry data from micro-dissected rat tubules were downloaded from Kidney Tubules Expression Atlas data table page (https://esbl.nhlbi.nih.gov/KTEA/). Kidney- and liver-enriched as well as group enriched genes were downloaded from the Human Protein Atlas (HPA, https://www.proteinatlas.org/humanproteome/tissue). The male WBM reconstruction can be downloaded from the virtual metabolic human database (https://www.vmh.life/#downloadview). Source data are provided with this paper.

## Code availability

Each use of software programs has been clearly indicated, and information on the options is provided in the Methods section. Source code to call on the listed software programs is available upon request. Both the COBRA toolbox and the PSCM toolbox can be obtained from github: https://opencobra.github.io/. The code for the simulations can be found at https://github.com/ThieleLab/CodeBase/tree/master/Cheng_et_al_IEM_simulations.

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

## Acknowledgements

The work of Y.C. was supported by the Chinese Scholarship Council. A.K. was supported by German Research Foundation (DFG) grant KO 3598/5-1 and DFG - Project-ID 431984000 - SFB 1453. Y.L. was supported by grant DFG KO 3598/4-1 and KO 3598/4-2 (to A.K.), and Pa.S. was supported by DFG CRC 992 (to A.K.) and the EQUIP Program for Medical Scientists, Faculty of Medicine, University of Freiburg. The work of M.S. was funded by the European Research Council (ERC Starting grant TREATCilia, grant No 716344 and by DFG - Project-ID 431984000 - SFB 1453). The work of J.H. and I.T. was supported by the European Research Council (ERC) under the European Union's Horizon 2020 research and innovation programme (grant agreement No 757922) to I.T. Genotyping was supported by Bayer Pharma AG. The GCKD study was/is funded by grants from the Federal Ministry of Education and Research (BMBF, grant number 01ER0804, K.U.E.) and the KfH Foundation for Preventive Medicine. We are grateful for the willingness of the patients to participate in the GCKD study. The enormous effort of the study personnel of the various regional centers is highly appreciated. We thank the large number of nephrologists who provide routine care for the patients and collaborate with the GCKD study. The GCKD Investigators are listed in Supplementary Note 1. This research has been conducted using the UK Biobank Resource (Application 28807).

## Author contributions

Design of this study: Y.C., Y.L., A.K. Recruitment and management of study: P.J.O., F.K., K.U.E., A.K. Bioinformatics, statistical analysis and computational modeling: Y.C., Pa.S., J.H., P.S., J.M., D.F.F., I.T., Y.L., A.K. Wrote the manuscript: Y.C., A.K. Critically read and approved the manuscript: Y.C., Pa.S., J.H., P.S., P.J.O., U.S., J.M., D.F.F., M.S., F.K., K.U.E., I.T., Y.L., A.K.

## Funding

## Competing interests

J.M. and D.F.F. are full time employees of Bayer AG, Division Pharma. All remaining authors declare no competing interests.

## Additional information

## GCKD Investigators

Peter J. Oefner[5], Florian Kronenberg[9], Kai-Uwe Eckardt[10,11] & Anna Köttgen[1,13]✉

A full list of members and their affiliations appears in the Supplementary Information.

