## [Peer Review File · Nature Communications]

Reviewer #1 (Remarks to the Author):

In the contribution "Rare genetic variants affect urine metabolite concentrations: linking population variation to inborn errors of metabolism and a virtual metabolic human", Cheng et al describe a gene based analyses of heritable component associated with human urine metabolite concentrations using data from 4864 participants. Albeit the manuscript is well written and study well conducted, it is missing a crucial component that would allow to assess the added value of the gene based tests over univariate standard GWAS approach. The cohort has been genotyped with the 2.5Omni-exome array, however, the authors decided to focus on the exome content and disregard the non-coding variation completely. I would find it important to contrast the GWAS approach with burden approach. As it stands, it is impossible to assess the added value of this contribution.

Reviewer #2 (Remarks to the Author):

Rare genetic variants affect urine metabolite concentrations: linking population variation to inborn errors of metabolism and a virtual metabolic human

Yurong Cheng, Pascal Schlosser, Johannes Hertel, Peggy Sekula, Peter J. Oefner, Ute 5 Spiekerkoetter, Johanna Mielke, Daniel F. Freitag, Miriam Schmidts, GCKD Investigators, Florian Kronenberg, Kai-Uwe Eckardt, Ines Thiele, Yong Li, Anna Köttgen

This study sought to detect additional components of human metabolism, to connect population studies to inborn errors of metabolisms, health outcomes, and the computational modeling of human metabolism to enable novel insights into metabolic diseases in humans.

This is a well conducted study with interesting findings, and I only have few minor suggestions.

Comment 1: Supplementary Figure 1 (analysis workflow): Add this figure to the main paper. It makes it much easier to understand the flow of numerous analyses and findings.

Comment 2: Of the 30 unique genes that were associated with either metabolite or metabolite ratios, 16 were previously linked with IEMs. Were the remaining 14 genes novel? In addition to replicating previous findings, the authors should highlight these potentially novel genes in their downstream analysis.

Comment 3: As authors state, previous rare variant association studies have been mostly conducted for metabolite levels in blood. Although plasma and urine metabolites can differ, it would be interesting to see a comparison of results between this study and e.g. Rhee et al. and Long et al.

Comment 4: Aggregate effect utilizing UKBB WES (50k individuals) testing 30 identified genes: There were no significant findings after multiple correction, although the sample size is quite large. Can the identified associations be attributed to CKD? Please elaborate this further in the discussion in the context of generalizability to population-based sample.

Reviewer #3 (Remarks to the Author):

Rare genetic variants affect urine metabolite concentrations: linking population variation to inborn errors of metabolism and a virtual metabolic human

Using an array approach the authors studied 226,233 genetic variants from up to 4,864 participants of the GCKD study. They combined this with a metabolomics approach (performed by Metabolon) for untargeted metabolomics evaluation of urine samples. The authors linked urinary

metabolite- and ratio values with genetic variants found in heterozygous individuals. Partly genes known from Inborn error of metabolism patients were found which mainly served the purpose of validating the methodology used. Also metabolites could be linked to genes that are not connected to known IEMs.

45 There were 128 significant associations (53 metabolite-gene and 75 metabolite ratio-gene pairs) involving 30 unique genes, 16 of which are known to underlie recessively inherited inborn errors of metabolism (IEMs).

When the authors evaluate urinary individual metabolites in this manuscript: do they normalise for creatinine? I do not see that in the method description. If the data are not corrected for creatinine or otherwise the influence of water intake will be substantial and it would be surprising to find 53 metabolite-gene associations.

Do the 75 metabolite ratio-gene pairs include creatinine ratio's like Phe/creatinine etc? Suggest to clarify this in the methods section.

163 In general, largest effect sizes of qualifying variants on metabolites were observed for variants with lower minor allele frequency, and stop gain and splice alleles tended to have larger effects than non-synonymous variants (Figure 1).

Effect on what? What if there was an effect on 2 metabolites? Do you show the metabolite-effect of the metabolite having the largest effect? Hard to see that in the figure..... I guess this needs some more explaining in the figure legend.

184 Figure 2A illustrates, however, that some of the aggregated variants based on these properties are likely to be neutral: for example, five of the six carriers of the missense variant encoded by rs143793815 (ASL p.T131M, NP_000039.2) had argininosuccinate levels below 0, while most carriers of the other aggregated variants showed clearly elevated levels well beyond the 75th percentile among controls.

Can you colour code the 6 individuals with heterozygous rs143793815 in fig 2A?

Legend Fig 2A: what is an Inverse normal transformed metabolite level? Explain what the vertical axis means. Also in the main text this is confusing: "... argininosuccinate levels below 0..." obviously does not refer to the concentration of the metabolite. Suggest to rephrase...

It helps when the fig 2 and the supplemental figs also show the significance levels of the various groups.

218 Typo: fractures of the pelvis

368 The fact that the respective genes can still be identified in our study via altered metabolite levels in heterozygous carriers underlines their validity as biomarkers of carrier status.

This seems an overstatement; in general a metabolite concentration will not perform remotely reliable as biomarker for the carrier status.

448 Number of different analytical columns and column type information is missing.

Tab 2 The table 2 in the main does not contain metabolite information on all genes given in figure 3. More detailed info is available in supplemental table 2. Suggest to complete the table 2.

Tab 2 What does the word "effect" in table 2 mean? Do effect values >1 indicate increased metabolite levels and <1 decreased levels? Is that increased compared to the mean of wild type individuals? If so an effect value 1.58 for 3-ureidopropionate in UPB1 defect indicates increased levels (which we would expect) and 0.54 for N-acetylglutamine a decreased level in ACY1 which would be rather unexpected?

Tab 2 There are several unknowns (unidentified features) in table 2. It gives the most relevant unidentified features for this study. It will be relevant to identify these features as they may carry important information of the function of the genes to which they seem to correlate. The authors

discuss the potential source of X-10457 in the manuscript. It is important to show the available additional MS/MS information on all unknowns in table 2 in the supplemental of this paper for other studies to build on. Also I would appreciate a few discussion lines on how the authors could further study these unknowns to in the end identify their molecular identity.

Discussion For each of the genes mentioned in figure 3 it would be helpful to have an evaluation of the metabolite effects that are found in table 2 and supplemental table 2. Is the effect (decreased or increased urinary concentration; pathway involved etc etc) in line with the expected metabolic effect based on the known function (if any) of the gene involved. And if not: which lessons do we learn?

Point-by-Point Response
NCOMMS-20-27631-T

We thank the Reviewers for taking the time to evaluate our manuscript, and for the many helpful and constructive comments. We have addressed them as outlined below, and believe that this has helped to substantially improve our work. The page and paragraph numbers below refer to the revised version of our manuscript, in which the corresponding changes are marked using track changes.

Reviewer #1 (Remarks to the Author):

In the contribution "Rare genetic variants affect urine metabolite concentrations: linking population variation to inborn errors of metabolism and a virtual metabolic human", Cheng et al describe a gene based analyses of heritable component associated with human urine metabolite concentrations using data from 4864 participants. Albeit the manuscript is well written and study well conducted, it is missing a crucial component that would allow to assess the added value of the gene based tests over univariate standard GWAS approach. The cohort has been genotyped with the 2.5Omni-exome array, however, the authors decided to focus on the exome content and disregard the non-coding variation completely. I would find it important to contrast the GWAS approach with burden approach. As it stands, it is impossible to assess the added value of this contribution.

Response: We agree with the Reviewer that it is interesting to evaluate the contribution of rare variants in the context of common genetic variation (typically assessed by standard GWAS). We had designed our study to focus on rare variants and gene-based tests only for two reasons: first, focusing on rare and presumably deleterious exonic variants for inclusion in the gene-based tests directly implicates the respective gene. This is important for the second part of our study, the *in silico* knock-out modeling of the implicated gene in the virtual metabolic human. The very high concordance between observed and predicted metabolite changes from our empirical population data and the computational modeling, respectively, supports our approach to use rare variant associations to implicate genes. In contrast, common variant associations studied in GWAS often implicate loci that contain many genes and also identify many regulatory variants, making the underlying gene difficult to identify. Secondly, we already published the results from GWAS of urinary metabolite associations for variants of minor allele frequency >1% in the GCKD study¹, and did not want to present duplicate information.

To address the Reviewer's important comment to which degree these two approaches overlap, we have now performed additional analyses. We first assessed the overlap between common (standard GWAS) and rare variant (gene-based tests) association signals as now outlined in the Methods (page 23, paragraph 2). When the associated metabolite was the same or closely related, we repeated the gene-based tests conditioned on the common GWAS index SNP. The results are summarized in a new Table 4: the gene-based test associations were stable for 10 of the 12 genes with common-variant signals in *cis* in conditional analyses (effect size changes within +/- 20%). Conversely, the effect of rare variant aggregate signals was abolished after conditioning on the respective common variant signal for two genes, *PYROXD2* and *RIOX1*. We have described this new information in the results section on page 7, paragraph 2.

Reviewer #2 (Remarks to the Author):

Rare genetic variants affect urine metabolite concentrations: linking population variation to inborn errors of metabolism and a virtual metabolic human

Yurong Cheng, Pascal Schlosser, Johannes Hertel, Peggy Sekula, Peter J. Oefner, Ute S. Pieper-Koetter, Johanna Mielke, Daniel F. Freitag, Miriam Schmidts, GCKD Investigators, Florian Kronenberg, Kai-Uwe Eckardt, Ines Thiele, Yong Li, Anna Köttgen

This study sought to detect additional components of human metabolism, to connect population studies to inborn errors of metabolisms, health outcomes, and the computational modeling of human metabolism to enable novel insights into metabolic diseases in humans.

This is a well conducted study with interesting findings, and I only have few minor suggestions.

Response: Thank you for the positive feedback.

Comment 1: Supplementary Figure 1 (analysis workflow): Add this figure to the main paper. It makes it much easier to understand the flow of numerous analyses and findings.

Response: We agree that the figure is helpful, and have now added it to the main manuscript as Figure 1. The numbering of the other main figures has been shifted.

Comment 2: Of the 30 unique genes that were associated with either metabolite or metabolite ratios, 16 were previously linked with IEMs. Were the remaining 14 genes novel? In addition to replicating previous findings, the authors should highlight these potentially novel genes in their downstream analysis.

Response: Yes, these 14 genes have not been linked to IEMs so far, and are not listed as causing a monogenic disease in OMIM. As outlined in the manuscript, these genes therefore represent suitable candidates for yet un-identified IEMs and can be tested when patients with abnormal values of the corresponding metabolites. To provide readers with a better overview, we have added a new Supplementary Table 8 that summarizes, for each of the 30 genes, the evidence from OMIM and the Human Metabolome, as well as an agreement with corresponding changes of the implicated metabolites in metabolic diseases (Metagene database) and in earlier rare variant studies of the serum or plasma metabolome. Several of the 16 genes known to underlie IEMs were also identified as associated with the corresponding metabolites in these earlier studies: *DMGDH*, *CTH*, *PAH*, and *UPBI*. Because these studies were not based on urine metabolite levels, we would not term this “replication”.

Selected showcases of the 14 genes not yet implicated in IEMs were already discussed, including *ALDH9A1* (page 10, paragraph 1). A systematic investigation of all 14 genes is challenging, because several of the genes have not been studied in great detail, and in many instances the associated metabolite was an unknown molecule. There are, however, some cases where our observations match existing biochemical knowledge. For instance, *AFMID* is

associated with the metabolites X-24455 and N-formylanthranilic acid. Based on the gene's known role in catalyzing the hydrolysis of N-formyl-L-kynurenine to L-kynurenine, and the knowledge that N-formylanthranilic acid can be synthesized from N-formyl-L-kynurenine, we would expect that loss of function variants would lead to higher levels of N-formyl-L-kynurenine and hence N-formylanthranilic acid. This is consistent with our observations. The association between *FOLH1* and N-acetyl-aspartyl-glutamate (NAAG) is another example where our observations match expectations based on biochemical knowledge.

Comment 3: As authors state, previous rare variant association studies have been mostly conducted for metabolite levels in blood. Although plasma and urine metabolites can differ, it would be interesting to see a comparison of results between this study and e.g. Rhee et al. and Long et al.

Response: We agree and have added this information to the new Supplementary Table 8, mentioned above. In addition to the studies by Rhee and Long, we identified two additional relevant publications. These four studies have been reviewed for associations of the genes identified in our study with plasma or serum levels of the corresponding metabolite(s), and five such associations were identified, with matching directions as indicated in the table. It needs to be noted, however, that the studies by Rhee *et al* and by Li *et al* used another platform for metabolite quantification, such that any signals that are not identified in studies of plasma metabolites could also be due to missing metabolites, as well as be affected by differences in power and measurement of genotypes (array vs. sequencing in Long *et al*). We have added a statement about the overlap with previous studies to the manuscript on page 15, paragraph 2.

Comment 4: Aggregate effect utilizing UKBB WES (50k individuals) testing 30 identified genes: There were no significant findings after multiple correction, although the sample size is quite large. Can the identified associations be attributed to CKD? Please elaborate this further in the discussion in the context of generalizability to population-based sample.

Response: We believe there may have been a misunderstanding. In our study, the outcome of interest were the levels of metabolites in urine. Metabolites have not yet been quantified in the UK Biobank, so that we did not test the same phenotype in the UK Biobank. Instead, we asked whether the same metabolite-associated rare variants that we identified in our study were associated with a clinical disease in the UK Biobank. If so, this could point towards an association with disease which may be mediated by the metabolite. We have discussed this for the case of the *ALPL* gene, phosphate-related metabolites in our study, and fractures in the UK Biobank on page 10, paragraph 2.

The observation that associations with molecular intermediate phenotypes such as metabolites are much stronger than with a complex disease that is associated with the intermediate phenotype is quite common. For instance, the association between genetic variants in the gene encoding HMG-CoA reductase with LDL cholesterol levels are much stronger than with cardiovascular disease. In addition, urine metabolite levels may not always reflect their levels in plasma, which can also impact the ability to observe associations with complex traits and diseases.

Lastly, we do not believe that the observed gene-metabolite associations are largely driven by the presence of CKD. First, half of the gene-metabolite pairs identified in our study is

supported by evidence from already known IEMs as summarized in the new Supplementary Table 8, many of which do not feature CKD. Second, some of the associations we identified have already been reported from plasma metabolome studies in general population-based samples. Third, we have previously systematically compared genetic association signals with urine metabolite levels between individuals with and without CKD, and found that genetic effects detected in CKD patients were well predicted those from individuals without CKD (model $R^2=0.95$)¹. As suggested, we have included a thorough discussion of this aspect on page 16, paragraph 3.

Reviewer #3 (Remarks to the Author):

Rare genetic variants affect urine metabolite concentrations: linking population variation to inborn errors of metabolism and a virtual metabolic human

Using an array approach the authors studied 226,233 genetic variants from up to 4,864 participants of the GCKD study. They combined this with a metabolomics approach (performed by Metabolon) for untargeted metabolomics evaluation of urine samples. The authors linked urinary metabolite- and ratio values with genetic variants found in heterozygous individuals. Partly genes known from Inborn error of metabolism patients were found which mainly served the purpose of validating the methodology used. Also metabolites could be linked to genes that are not connected to known IEMs.

45 There were 128 significant associations (53 metabolite-gene and 75 metabolite ratio-gene pairs) involving 30 unique genes, 16 of which are known to underlie recessively inherited inborn errors of metabolism (IEMs).

When the authors evaluate urinary individual metabolites in this manuscript: do they normalise for creatinine? I do not see that in the method description. If the data are not corrected for creatinine or otherwise the influence of water intake will be substantial and it would be surprising to find 53 metabolite-gene associations.

Do the 75 metabolite ratio-gene pairs include creatinine ratio's like Phe/creatinine etc? Suggest to clarify this in the methods section.

Response: Yes, the metabolite concentrations have been corrected for urine dilution using pq-normalization, as detailed in the Methods section on page 21, paragraph 3. Correction based on the concentrations of many metabolites with complete information, as done in pq-normalization, has been shown to be preferable over other methods to normalize dilution². We have previously published that indeed pq-normalization as compared to creatinine-normalization led to better power for the identification of known genetic associations with urine metabolite concentrations³. Ratios including creatinine were not among the 75 gene-metabolite pairs (all shown in Supplementary Table 4).

163 In general, largest effect sizes of qualifying variants on metabolites were observed for variants with lower minor allele frequency, and stop gain and splice alleles tended to have larger effects than non-synonymous variants (Figure 1).

Effect on what? What if there was an effect on 2 metabolites? Do you show the metabolite-effect of the metabolite having the largest effect? Hard to see that in the figure..... I guess this needs some more explaining in the figure legend.

Response: Figure 2 (formerly Figure 1) contains a symbol for the effect of each variant that contributed to an aggregate association signal on the level of each of the implicated metabolites. Variants that show associations with more than one metabolite are therefore represented more than once, because their effects on the implicated metabolites can differ. To make data comparable across metabolites, we chose a rank-based inverse normal transformation. Thus, the units of the Y axis represent one standard deviation. To give a comprehensive overview of all findings and avoid incorrect impressions by only showing the metabolite on which a variant had the largest effect, all metabolite-variant associations have been included in Figure 2. As suggested by the Reviewer, we have now clarified this in the figure legend.

184 Figure 2A illustrates, however, that some of the aggregated variants based on these properties are likely to be neutral: for example, five of the six carriers of the missense variant encoded by rs143793815 (ASL p.T131M, NP_000039.2) had argininosuccinate levels below 0, while most carriers of the other aggregated variants showed clearly elevated levels well beyond the 75th percentile among controls.

Can you colour code the 6 individuals with heterozygous rs143793815 in fig 2A?

Response: Thank you for this helpful suggestion, which we have implemented in an updated figure (now numbered 3A).

Legend Fig 2A: what is an Inverse normal transformed metabolite level? Explain what the vertical axis means. Also in the main text this is confusing: “.. argininosuccinate levels below 0...” obviously does not refer to the concentration of the metabolite. Suggest to rephrase...

Response: We have now clarified that the metabolite levels were transformed (rank-based inverse normal transformation) to achieve a normal distribution and to allow for comparisons across metabolites. We have updated the figure legend (now Figure 3A) to better explain this, and have replaced levels “below 0” with levels “below the median” in the manuscript (starting on page 8, paragraph 2). Thank you for pointing this out.

It helps when the fig 2 and the supplemental figs also show the significance levels of the various groups.

Response: We agree that this is helpful information, which we had previously included in Supplementary Tables 2 (single metabolites) and 5 (ratios). We have now updated Figure 3 (the former Figure 2) and Supplementary Figure 1 (former SF2) to include the variant consequence.

218 Typo: fractures of the pelvis

Response: Thank you, this has been fixed.

368 The fact that the respective genes can still be identified in our study via altered metabolite levels in heterozygous carriers underlines their validity as biomarkers of carrier status.

This seems an overstatement; in general a metabolite concentration will not perform remotely reliable as biomarker for the carrier status.

Response: We agree with the Reviewer and have rephrased this sentence as follows: “In general, heterozygous carriers of causative mutations for IEMs are not clinically affected, suggesting that residual function of the encoded proteins is sufficient to perform their physiological functions. However, the resulting changes in metabolite levels were still sufficient to identify the respective genes in our study.” (page 17, paragraph 1).

448 Number of different analytical columns and column type information is missing.

Response: We have now added this information, along with a more detailed description of the methods for metabolite quantification, to a new section of the Supplementary Information (Supplementary Note 2 in the separate Supplementary Information document).

Tab 2 The table 2 in the main does not contain metabolite information on all genes given in figure 3. More detailed info is available in supplemental table 2. Suggest to complete the table 2.

Response: The apparent difference results from the fact that the former Figure 3 (updated Figure 4) contains all genes identified in association with single metabolites (shown in Table 2) as well as genes identified in association with metabolite ratios (shown in Table 3). Across Tables 2 and 3, full metabolite information on all genes is given. The more detailed information in Supplementary Table 2 (single metabolites) and Supplementary Table 5 (metabolite ratios) shows information at the level of the individual variants that were aggregated for the gene-based tests, whereas the main tables show the aggregated effects of these variants on metabolite levels.

Tab 2 What does the word “effect” in table 2 mean? Do effect values >1 indicate increased metabolite levels and <1 decreased levels? Is that increased compared to the mean of wild type individuals? If so an effect value 1.58 for 3-ureidopropionate in UPB1 defect indicates increased levels (which we would expect) and 0.54 for N-acetylglutamine a decreased level in ACY1 which would be rather unexpected?

Response: “Effect” denotes the cumulative impact of each copy of rare, potentially damaging variants on metabolite levels, i.e. the average effect for each copy of a rare variant that an individual carries. For example, urine 3-ureidopropionate levels among heterozygous carriers of one rare *UPBI* variant were an estimated 1.58 standard deviations higher than those of individuals who did not carry any such variants. This is consistent with loss-of-function as a mechanism and matches clinical observations from the known corresponding IEM. For N-acetylglutamine, it means that the average urine levels were 0.54 standard deviations higher per

variant copy among carriers as compared to non-carriers, which again is consistent with expectation. To clarify this, we have now added an explanation to the footnote of Table 2.

Tab 2 There are several unknowns (unidentified features) in table 2. It gives the most relevant unidentified features for this study. It will be relevant to identify these features as they may carry important information of the function of the genes to which they seem to correlate. The authors discuss the potential source of X-10457 in the manuscript. It is important to show the available additional MS/MS information on all unknowns in table 2 in the supplemental of this paper for other studies to build on. Also I would appreciate a few discussion lines on how the authors could further study these unknowns to in the end identify their molecular identity.

Response: We agree with the Reviewer that de-orphanization of the unknown molecules has the potential to improve our understanding of the functions of the associated gene. We had therefore already included information on mass-to-charge ratio, retention time, and platform in Supplementary Table 1. Metabolite levels were quantified as a service at Metabolon *Inc.*, and this information is returned to customers. The mass spectra themselves are proprietary and not accessible to us. However, we have negotiated the deposition of mzML files to the EMBL-EBI database Metabolights (<https://www.ebi.ac.uk/metabolights/>) with the company. The uploaded samples include the unknown molecules in Table 2, and can be obtained from there. We did not obtain the accession number yet, but will add this to the manuscript prior to publication.

Lastly, we have expanded the discussion to outline ways to potentially de-orphanize the unknown molecules' identities (page 19, paragraph 1), for example through unsupervised clustering with other metabolites of known identity. Together with the information on m/z and retention time, this can greatly reduce the search space. In fact, we have previously used such an approach to successfully de-orphanize X-13689 as the glucuronide of α -CMBHC, consistent with the role of the implicated gene in Vitamin E metabolism¹.

Discussion For each of the genes mentioned in figure 3 it would be helpful to have an evaluation of the metabolite effects that are found in table 2 and supplemental table 2. Is the effect (decreased or increased urinary concentration; pathway involved etc etc) in line with the expected metabolic effect based on the known function (if any) of the gene involved. And if not: which lessons do we learn?

Response: We agree that this is relevant and have added information to the new Supplementary Table 8. In each case where an IEM is known for a given gene and the implicated metabolite has been reported from the affected patients, the expected direction of metabolite changes matches the one we observe (column J). In addition, the direction matches with known biochemical knowledge about encoded enzymes' functions, assuming loss-of-function as a mechanism. For the genes that have not yet been implicated in an IEM, it is more difficult to provide information. Several of the genes have not been studied in great detail, and in many instances the associated metabolite is an unknown molecule. There are, however, some cases where our observations match existing biochemical knowledge. For instance, *AFMID* is associated with the metabolites X-24455 and N-formylanthranilic acid. Based on the gene's known role in catalyzing the hydrolysis of N-formyl-L-kynurenine to L-kynurenine, and the knowledge that N-formylanthranilic acid can be synthesized from N-formyl-L-kynurenine, we would expect that

loss of function of *AFMID* would lead to higher levels of N-formyl-L-kynurenine and hence N-formylanthranilic acid. This is consistent with our observations. The association between *FOLH1* and N-acetyl-aspartyl-glutamate (NAAG) is another example where our observations match expectations based on biochemical knowledge.

References

1. Schlosser P, *et al.* Genetic studies of urinary metabolites illuminate mechanisms of detoxification and excretion in humans. *Nat Genet* **52**, 167-176 (2020).
2. Dieterle F, Ross A, Schlotterbeck G, Senn H. Probabilistic quotient normalization as robust method to account for dilution of complex biological mixtures. Application in 1H NMR metabonomics. *Anal Chem* **78**, 4281-4290 (2006).
3. Li Y, *et al.* Genome-Wide Association Studies of Metabolites in Patients with CKD Identify Multiple Loci and Illuminate Tubular Transport Mechanisms. *J Am Soc Nephrol* **29**, 1513-1524 (2018).

Reviewer #1 (Remarks to the Author):

My major concern has been now adequately addressed and the results look very convincing. I have no further comments to add.

Reviewer #2 (Remarks to the Author):

All my questions/concerns have been addressed in the revisions. No further questions.

Reviewer #3 (Remarks to the Author):

Two points remaining:

1. Give the specifics for the column used for "basic negative ion conditions - a separate dedicated C18 column".
2. Suggest to add the URL and the accession number to enable finding the uploaded data in the manuscript ("However, we have negotiated the deposition of mzML files to the EMBL-EBI database Metabolights (<https://www.ebi.ac.uk/metabolights/>) with the company. The uploaded samples include the unknown molecules in Table 2, and can be obtained from there. We did not obtain the accession number yet, but will add this to the manuscript prior to publication.")

Reviewer #3 (Remarks to the Author):

1. Give the specifics for the column used for "basic negative ion conditions - a separate dedicated C18 column".

Response: We have now clarified in the Supplementary Information (page 20) that a separate C18 column of the same kind (Waters UPLC BEH C18-2.1x100 mm, 1.7 μm) was used: "basic negative ion conditions: this optimized method used a separate dedicated C18 column (Waters UPLC BEH C18-2.1x100 mm, 1.7 μm)".

2. Suggest to add the URL and the accession number to enable finding the uploaded data in the manuscript ("However, we have negotiated the deposition of mzML files to the EMBL-EBI database Metabolights (<https://www.ebi.ac.uk/metabolights/>) with the company. The uploaded samples include the unknown molecules in Table 2, and can be obtained from there. We did not obtain the accession number yet, but will add this to the manuscript prior to publication.")

Response: The Metabolights ID of our study is MTBLS284. A temporary URL to pre-view the study is <https://www.ebi.ac.uk/metabolights/reviewercc851652b3101362d5055c056a9f74e1>. The formal URL will be <http://www.ebi.ac.uk/metabolights/MTBLS284>. We have included the Metabolights ID and the URL into the Data Availability section of the manuscript (page 29).